# Holocentromeres can consist of merely a few megabase-sized satellite arrays

Yi-Tzu Kuo ®[1] ✉, Amanda Souza Câmara[1], Veit Schubert ®[1], Pavel Neumann ®[2], Jiří Macas ®[2], Michael Melzer[1], Jianyong Chen[1], Jörg Fuchs ®[1], Simone Abel[3], Evelyn Klocke ®[3], Bruno Huettel ®[4], Axel Himmelbach[1], Dmitri Demidov[1], Frank Dunemann ®[3], Martin Mascher ®[1], Takayoshi Ishii ®[5], André Marques ®[6] & Andreas Houben ®[1] ✉

The centromere is the chromosome region where microtubules attach during cell division. In contrast to monocentric chromosomes with one centromere, holocentric species usually distribute hundreds of centromere units along the entire chromatid. We assembled the chromosome-scale reference genome and analyzed the holocentromere and (epi)genome organization of the lilioid *Chionographis japonica*. Remarkably, each of its holocentric chromatids consists of only 7 to 11 evenly spaced megabase-sized centromere-specific histone H3-positive units. These units contain satellite arrays of 23 and 28 bp-long monomers capable of forming palindromic structures. Like monocentric species, *C. japonica* forms clustered centromeres in chromocenters at interphase. In addition, the large-scale eu- and heterochromatin arrangement differs between *C. japonica* and other known holocentric species. Finally, using polymer simulations, we model the formation of prometaphase line-like holocentromeres from interphase centromere clusters. Our findings broaden the knowledge about centromere diversity, showing that holocentricity is not restricted to species with numerous and small centromere units.

The centromere is a specialized chromosome region where the kinetochore complex assembles, and spindle microtubules attach to ensure chromosome segregation during mitosis and meiosis. The chromosomal localization of the centromere is generally epigenetically marked by nucleosomes containing the centromere-specific histone H3 (CENH3, also called CENP-A). The length of centromeres ranges from as small as 120 bp to up to several megabases of DNA (reviewed by[1]). Most studied species possess a single size-restricted centromere, the monocentromere, visualized as the primary constriction. In addition, holocentric (also termed holokinetic) species exist with centromeres distributed along the entire chromosome length[1].

Aside from its role in chromosome segregation, the centromere also plays a vital role in determining the large-scale genome architecture and chromatin composition (Muller et al.[2]). In contrast to most monocentric species, the higher-order organization of centromeres in holokinetic species like the nematode *Caenorhabditis elegans*[3] and plant species, the Juncaceae *Luzula elegans*[4] and the Cyperaceae *Rhynchospora pubera*[5], differ between interphase and metaphase. During interphase, holocentromeres are dispersed into many small centromeric units evenly distributed within the nucleus. At the onset of chromosome condensation, the centromeric units join and form line-like structures along chromatids. Due to this multi-centromere subunit structure, holocentric chromosomes could also be considered

[1]Leibniz Institute of Plant Genetics and Crop Plant Research (IPK) Gatersleben, Corrensstrasse 3, 06466 Seeland, Germany. [2]Biology Centre, Czech Academy of Sciences, Institute of Plant Molecular Biology, Branišovská 31, České Budějovice CZ-37005, Czech Republic. [3]Julius Kühn-Institute (JKI), Institute for Breeding Research on Horticultural Crops, Erwin-Baur-Str. 27, 06484 Quedlinburg, Germany. [4]Max Planck Genome-Centre Cologne, Max Planck Institute for Plant Breeding Research, Cologne 50829, Germany. [5]Arid Land Research Center, Tottori University, 1390 Hamasaka, Tottori 680-0001, Japan. [6]Department of Chromosome Biology, Max Planck Institute for Plant Breeding Research, Cologne 50829, Germany. ✉e-mail: kuo@ipk-gatersleben.de; houben@ipk-gatersleben.de

as 'polycentric'[6]. Polymer simulation suggests that the cell cycle-dependent assembly of the holocentromere relies on the interaction between centromeric nucleosomes and the structural maintenance of chromosomes (SMC) proteins[7].

Because holocentric taxa are often embedded within broad phylogenetic lineages possessing monocentric chromosomes, holocentric chromosomes are considered to be derived from monocentric ones. This transition occurred independently at least 13 times in distant lineages, including green algae, protozoans, invertebrates, as well as flowering plant families[8]. The factors that triggered this centromere-type transition and its mechanisms are currently unknown. Besides other models, a spreading of centromeric sequences from one location to multiple sites along the chromosomes has been proposed as a mechanism of holocentromere formation[9]. The existence of metapolycentric species possessing an elongated primary constriction containing multiple repeat-enriched centromeres supports this hypothesis[10,11].

The different types of holocentromeres likely depend on the organization of the monocentric precursor centromere and the evolutionarily developmental stage of the holocentromere[12]. Despite the importance of CENH3 in centromere identity, in four lineages of insects, the transition to holocentricity was associated with the loss of CENH3[13]. In other holocentrics, like *Meloidogyne* nematodes[14] and the plant *Cuscuta europaea*[15] the *CENH3* gene was duplicated. However, CENH3 probably lost its function in *Cuscuta* holocentrics[15]. Also, holocentric centromeres with and without centromere-specific repeats exist. In the CENH3-deficient moth *Bombyx mori* and the CENH3-possessing nematodes *C. elegans* and *Ascaris suum*, kinetochores assemble anywhere without sequence specificity along the chromosomes where nucleosome turnover is low[16]. On the other hand, holocentric chromosomes with centromere-specific repeats exist, e.g. in *R. pubera*[17] and the nematode *Meloidogyne*[14]. The genome of *R. pubera* harbors thousands of regularly spaced 15–25 kb-long CENH3-interacting satellite arrays underlying its holocentromeres[5,17]. Thus, the quandary between the exclusively epigenetic centromere definition and the role of centromeric DNA in mediating centromere identity is still unresolved[1].

To broaden our knowledge about the organization and diversity of the independently evolved holocentromeres and their interplay with the large-scale genome architecture and chromatin composition, we resolved the centromere and (epi)genome organization of the plant *Chionographis japonica*. The genus *Chionographis* belonging to the family Melanthiaceae is the only lilioid monocot known to include holokinetic species[18,19]. The holocentricity of *Chionographis* chromosomes was concluded based on the stable mitotic behavior of X-irradiation-induced chromosome fragments and the parallel separation of sister chromatids at anaphase[18,20]. However, the organization of this independently evolved holocentromere has not been characterized at the molecular level. Here we report a holocentromere composed of only a few evenly spaced CENH3-positive megabase pair-long satellite DNA arrays. We further reveal that the epigenetic regulation of repeat-based centromeres in monocentric and holocentric species is evolutionarily conserved. Using polymer simulations, we model the transition of *Chionographis* holocentromeres from interphase to prophase and discuss possible mechanisms driving the evolution of a repeat-based holocentromere.

## Results

### The holocentromere of *C. japonica* is CENH3-based and clusters during interphase near the nuclear membrane

Holocentric species with or without CENH3-based centromeres exist[13]. To test whether *C. japonica* (Supplementary Fig. 1) is a CENH3-possessing holocentric species, the root, flower, fruit, and leaf transcriptomes of this species were searched for *CENH3* transcripts. One *CENH3* gene was identified in all transcriptome datasets. The specificity of the generated anti-CENH3 antibody was confirmed by detecting the

predicted 18-kDa protein by Western blot analysis (Supplementary Fig. 2). Immunostaining and telomere-FISH of chromosomes revealed CENH3 signals distributed at metaphase on poleward surfaces (Fig. 1a), from telomere to telomere (Fig. 1b). Colocalization of CENH3 and spindle microtubule attachment sites along entire chromosomes further confirmed holocentricity (Fig. 1c, d, Supplementary Movie 1). But in contrast to the holocentric plants *L. elegans* and *R. pubera*, where the CENH3-positive centromere forms a longitudinal groove at metaphase[5,21], the centromere in *C. japonica* did not show such a structure (Supplementary Movie 2).

During the mitotic cell cycle, the line-like CENH3 signals appeared before the breakdown of the nuclear membrane at prophase (Supplementary Fig. 3). At late telophase, this line-like CENH3 distribution diverged (Supplementary Fig. 3e). Notably, at interphase, unlike other holocentric species, CENH3 signals clustered in distinct chromocenters (Fig. 1e, Supplementary Fig. 3f). These heterochromatic regions accumulate preferentially near the nuclear membrane, as demonstrated by transmission electron microscopy (Fig. 1f). A similar preference exists for the monocentromeres of *Arabidopsis thaliana* (Supplementary Fig. 4a)[22]. In contrast, holocentric species with many centromere units, such as *R. pubera*[5] and *L. elegans*[4], have many small heterochromatin regions without nuclear membrane association (Supplementary Fig. 4b–c). Thus, the centromere organization at interphase differs between *C. japonica* and other known holocentric species.

Prompted by the holocentromere-atypical interphase distribution of CENH3-positive chromocenters (Fig. 1e), we next investigated the number of CENH3 clusters in flow-sorted root G1 nuclei of *C. japonica*. An average of 68.17 and 67.42 signal clusters per nucleus, equivalent to 2.85 and 2.81 per chromatid, were counted in 2D and 3D images, respectively (Fig. 1g, Supplementary Table 1). Thus, considering a diploid chromosome number of 24[18], *C. japonica* forms, on average, as few as 2.8 CENH3-positive chromocenters per chromosome.

### *C. japonica* reveals a chromosome-wide distribution of kinetochore proteins and cell cycle-dependent histone marks

To further confirm the holocentricity of *C. japonica*, the distributions of MIS12 and NDC80, two conserved representative proteins of the outer kinetochore[23,24], were determined by anti-*C. japonica* MIS12 and NDC80 antibodies. Their immuno-signals revealed a distribution pattern similar to that for CENH3 throughout mitosis (Fig. 2a, b and Supplementary Fig. 5), confirming the holocentromeric nature of *C. japonica* chromosomes.

Next, we examined the histone phosphorylation marks, typically enriched at inner centromeres. The inner centromere is usually marked by phosphorylation of histone H3 threonine 3 (H3T3ph) and histone H2A threonine 120 (H2AT120ph) at metaphase[25], and the pericentromere is enriched in phosphorylated H3S10 and H3S28. Like in other holocentric species[26], in *C. japonica*, the H3S10ph signals were observed throughout mitotic metaphase chromosomes (Fig. 2c). Notably, both H3S28 and H3T3 hyperphosphorylation were mostly enriched in the inner centromere along the entire chromosomes (Fig. 2d–f). None of the phosphorylated histone marks displayed interphase signals. H2AT120ph, a highly conserved (peri)centromeric histone modification in plant species[27], was not detectable, suggesting that this type of phosphorylation was lost in *C. japonica*, or the histone H2A sequence altered in this species (Supplementary Fig. 6).

### The holocentromere of *C. japonica* is composed of a few evenly spaced megabase-scale satellite array-based centromere units

Prompted by the unusual holocentromere organization, we resolved the centromere and genome organization of *C. japonica* (2n = 24). First, we determined a genome size of 1C = 1368 Mb, and assembled a chromosome-scale reference genome sequence by integrating PacBio HiFi reads (58.5× genome coverage) and a Hi-C chromatin interaction

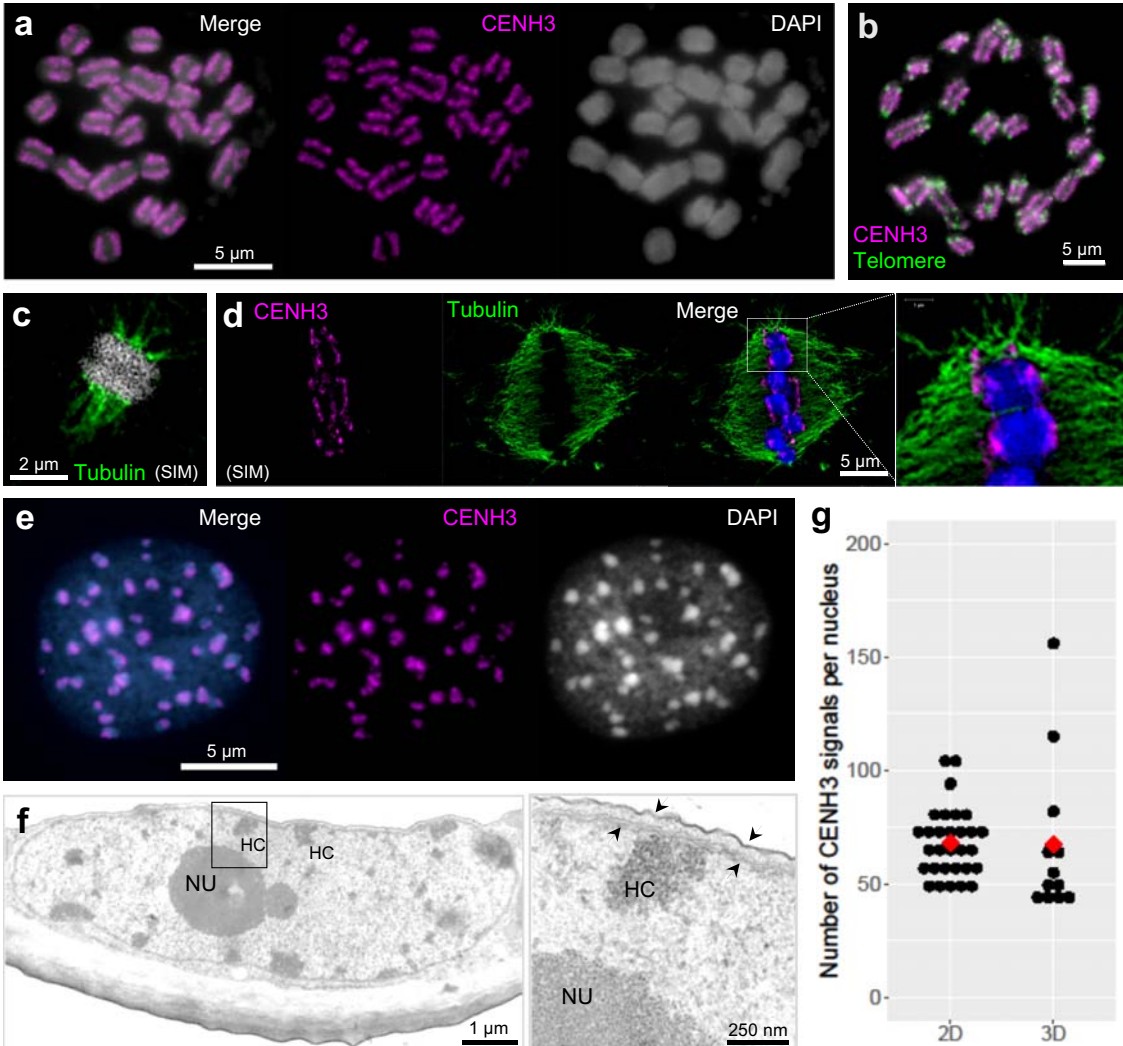

**Fig. 1 | *C. japonica* centromeres are distributed along entire mitotic chromosomes and form nuclear chromocenters. a** Condensed metaphase chromosomes show line-like CENH3 immuno-signals on the poleward surface of each chromatid, **b** from telomere to telomere. **c** Microtubules attach to the poleward surface of both chromatids. **d** Localization of CENH3 and tubulin sites. The enlargement shows the colocalization between CENH3 and microtubules. **e** CENH3 signals cluster in chromocenters of the interphase nucleus. **c**, **d** were taken by super-resolution microscopy (SIM). Chromosomes and nuclei were counterstained with DAPI.

**f** Transmission electron micrograph of a *C. japonica* interphase nucleus. Electron-dense heterochromatic chromocenters (HC) are often located in the proximity of the double-layered nuclear membrane (further enlarged insert, arrows). NU, nucleolus. **g** The number of CENH3 signal clusters per interphase nucleus counted in 2D (*n* = 30) and 3D (*n* = 12) image stacks. Red dots show the average number. **a**–**f** At least two independent experiments were carried out to confirm the reproducibility of the labeling patterns. **g** Source data are provided as a Source Data file.

dataset. The assembled genome sequence is that of an individual plant that had been clonally tissue cultured to harvest enough tissue for DNA extraction. The primary de novo genome assembly of *C. japonica* has 3786 contigs totaling 1,526,137,861 bp with a GC content of 41.26%, N50 of 2.88 Mb, and a complete BUSCO of 91.90% (Supplementary Table 2, Supplementary Fig. 7). After Hi-C scaffolding, 12 chromosome scaffolds were constructed, representing a total of 1090.73 Mb (N50 = 81.11 Mb), equivalent to ~80% of the *C. japonica* genome (Supplementary Fig. 8; Supplementary Table 3).

To address whether holocentromeres of *C. japonica* are repeat-based, we performed CENH3-ChIP sequencing and analyzed the read enrichment by an assembly-independent strategy using ChIP-Seq Mapper[28]. In the top 200 most abundant repeat clusters, clusters CL1 and CL73 revealed 11.1- and 11.7-fold enrichment in the immunoprecipitated fraction, respectively (Fig. 3a). CL1 is the most abundant repeat cluster (16.11%) in the genome. The two variants of CL1, named Chio1 and Chio2, are 23- and 28-bp monomer satellite, respectively (Fig. 3b). The consensus sequence of the dominant Chio1 contains a 5-bp

deletion and 1-bp substitution relative to that of the less abundant Chio2. CL73 is a Chio1/2-containing higher-order repeat cluster. The origin of the Chio repeats remains enigmatic, as they showed no similarity with any other known sequences. Notably, the CENH3-ChIP enriched sites coincided with the position of Chio1/2 satellite arrays in the assembled genome using the multi- and uni-mapping modes (Fig. 4a, Supplementary Fig. 9). Thus, the holocentromere of *C. japonica* is Chio satellite repeat-based.

Mapping the centromeric Chio1/2 satellite repeats and CENH3-ChIPseq reads on the 12 assembled chromosomes identified, on average, 8.3 centromere units with an average size of 1.89 Mb (ranging from 0.24 up to 4.46 Mb) per chromosome (Fig. 3c, d, Supplementary Table 3). Thus, on average, ~3 centromere units are present in each chromocenter at interphase. The amount of centromeric DNA is exceptional because ~17% of the genome contains CENH3-interacting DNA. All 12 assembled chromosomes of *C. japonica* contain, in total, 100 centromere units (Fig. 3c, Supplementary Table 3). The centromere units are relatively even-spaced, with an average interval

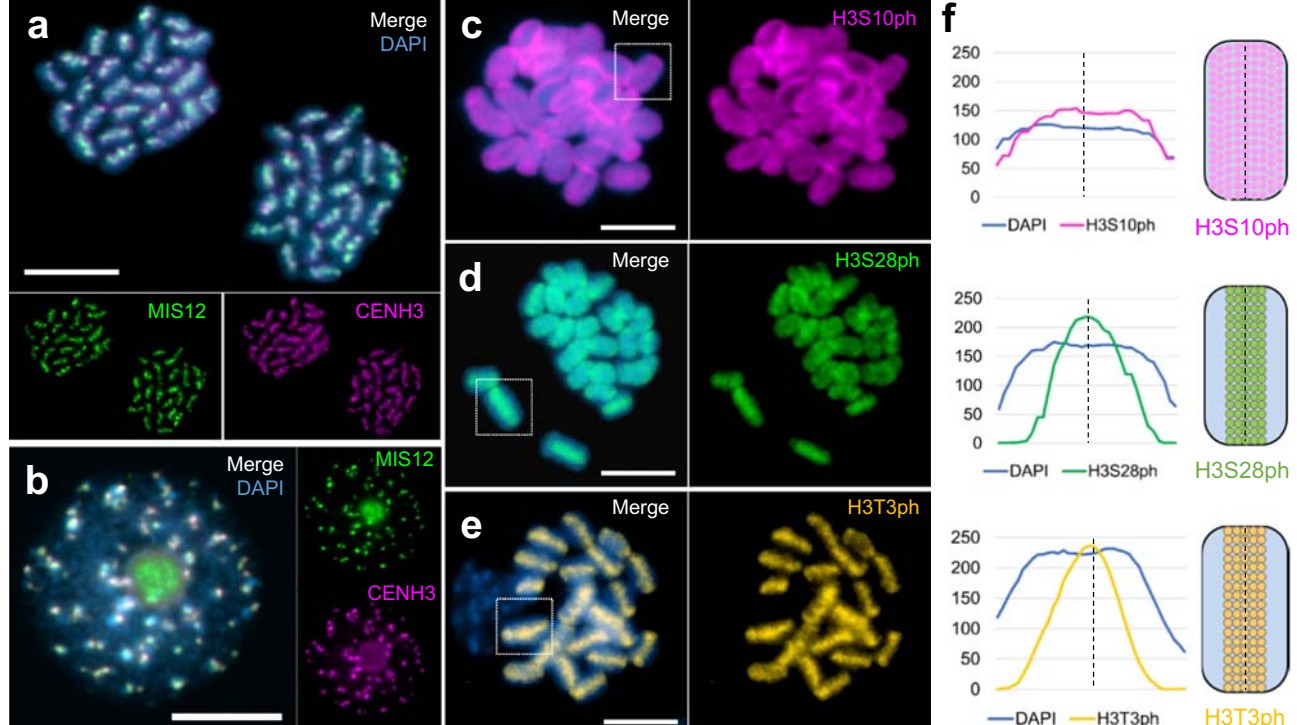

**Fig. 2 | *C. japonica* reveals a chromosome-wide distribution of kinetochore proteins and cell cycle-dependent histone marks.** Immunolabelling shows colocalization of the kinetochore protein MIS12 and CENH3 in (**a**) mitotic telophase chromosomes and (**b**) an interphase nucleus. Immuno-signals of the histone mark (**c**) H3S10ph (purple) show uniform labeling of mitotic metaphase chromosomes. The signals of (**d**) H3S28ph (green) and (**e**) H3T3ph (yellow) locate along the entire metaphase chromosomes where sister chromatids attach. Chromosomes were counterstained with DAPI. **f** Line scan plot profiles of individual chromosomes (**c**–**e**, squares) show the signal intensity of the three histone marks and corresponding DAPI-stained chromosomes. Immuno-signal distribution along single chromosomes is depicted as schemata next to the profiles. Scale bar, 5 μm. **a**–**e** At least two independent experiments were carried out to confirm the reproducibility of the labeling patterns. **f** Source data are provided as a Source Data file.

of 9.97 Mb between two adjacent centromere units (Supplementary Table 3). The sizes of centromere units and their flanking intercentromeric regions are weakly correlated (correlation coefficient = 0.21) (Supplementary Fig. 10).

Each centromere unit is composed of long tracks of Chio1/2 satellite arrays and is characterized by an individual mixture of forward- and reverse-oriented satellite arrays (Fig. 3e). Further, Chio1 and Chio2 repeat monomers contain three and four dyad symmetries, respectively (Fig. 3b). Particularly, a conserved 8-bp dyad symmetry was predicted to form a stable secondary hairpin loop structure between neighboring monomers (Fig. 3f). Whether the 8-bp sequence and hairpin loop structure are crucial for centromere identity is unknown.

The polycentromere-like genome organization suggested by our chromosome-scale sequence assembly was confirmed by Immuno-FISH. Naturally extended pachytene chromosomes showed, on average, nine evenly spaced distinct scattered CENH3- and Chio1-positive centromere units colocalizing with knob-like chromatin structures per chromosome (Fig. 3g, Supplementary Fig. 11a−c). In contrast, condensed mitotic metaphase chromosomes displayed Immuno- and FISH signals at poleward peripheries reminiscent of railroad tracks (Fig. 3h). Using super-resolution microscopy with a resolution of 120 nm indicated a ~65% overlap of Chio1 and CENH3 signals in interphase nuclei (Supplementary Fig. 11d). Thus, holocentromeres can be also formed by few evenly spaced CENH3-positive megabase-scale centromere units.

In addition to the centromeric Chio satellites, non-centromeric satellites like CjSat3, CjSat4, and CjSat5 displayed on metaphase chromosomes clustered, dispersed, or subtelomeric signals, respectively (Fig. 4a−d, Supplementary Table 4). The genome-wide domain-based annotation of transposable elements in *C. japonica* showed that, generally, a uniform distribution of both *Ty3/gypsy* and *Ty1/copia* retroelements in intercentromeric regions (Fig. 4a). The 45S ribosomal DNA is located on one chromosome pair in distal position (Fig. 4e), as typical for holocentric species[29].

## The *C. japonica* genome is organized in distinct chromosomal eu- and heterochromatic domains

Eu- and heterochromatin are interspersed in holocentric species with many small centromere units[5,17,30], while in small-genome monocentric species, both chromatin types form distinct chromosomal subdomains[31]. Methylation of lysine 9 of histone H3 (H3K9) is typical state for heterochromatin in pericentromeric regions of monocentromeres[32]. To determine whether holocentromeres which are based on a few megabase-sized satellite-DNA arrays affect the large-scale genome organization, the patterns of evolutionarily conserved eu- and heterochromatin-specific histone marks H3K4me2 and H3K9me2 were resolved at basepair-resolution by ChIP-seq in *C. japonica* using the multi- and uni-mapping modes (Supplementary Fig. 9).

Generally, CENH3-positive centromere units were H3K4me2-negative and flanked by H3K9me2 enriched regions (Fig. 5a). Parts of the centromere units were depleted of H3K9me2. Subtelomeric CjSat5 arrays were strongly associated with H3K9me2 (Supplementary Fig. 12). The intercentromeric regions were enriched in H3K4me2 and harbored the majority of mapped RNAseq reads matching the distribution of coding sequences (Fig. 5a, Supplementary Fig. 12). Locally, H3K4me2 was highly enriched at the promoter and terminal regions of genes, contrary to H3K9me2, and reduced in the centromeric regions (Fig. 5b). Thus, the genome-wide presence of CENH3-bound chromatin is positively correlated to H3K9me2 ($r = 0.33$) and negatively correlated to H3K4me2 ($r = -0.51$) and transcriptome ($r = -0.50$) (Fig. 5c).

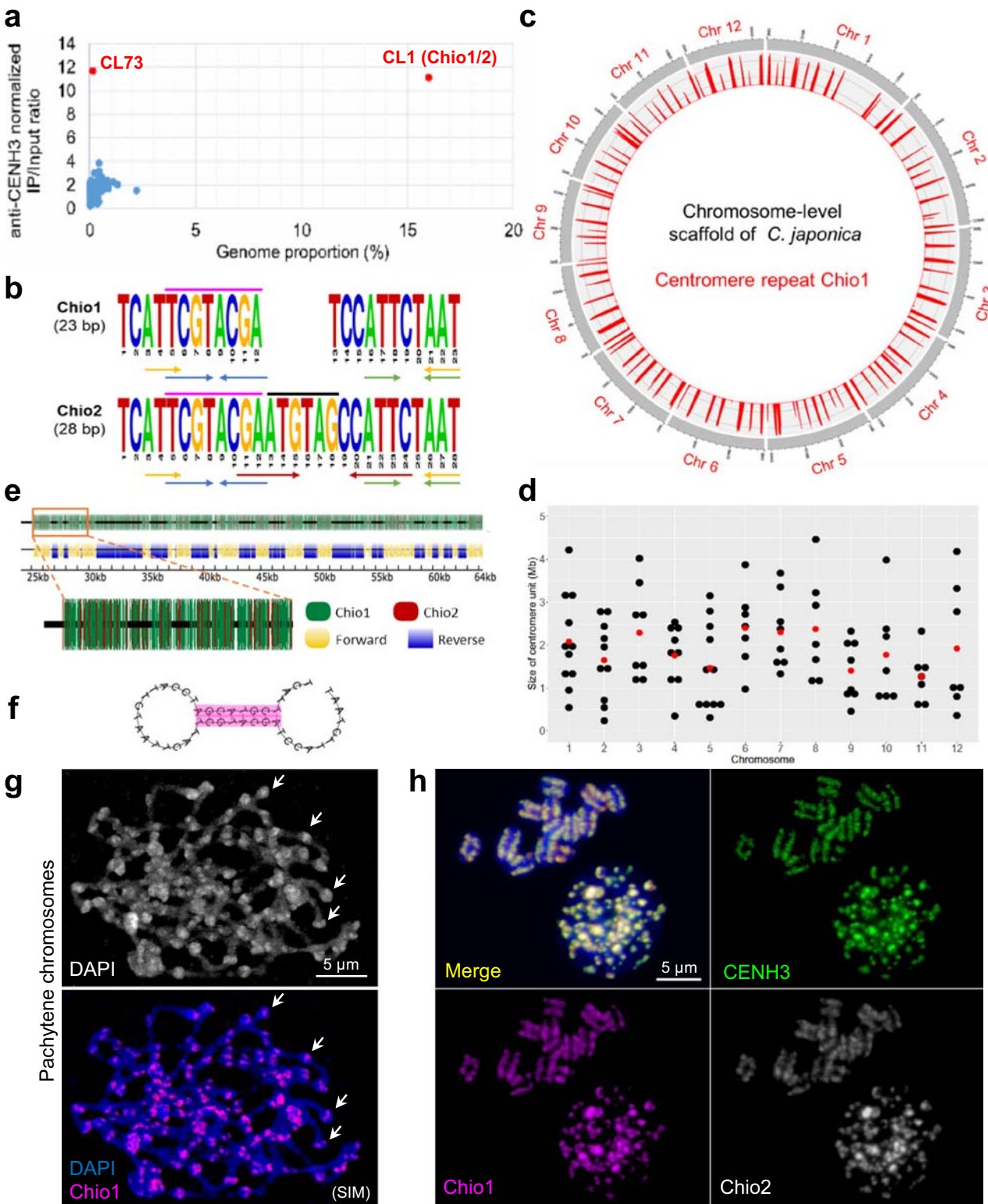

**Fig. 3 | The holocentromere of *C. japonica* is satellite repeat-based. a** The genome proportion and normalized enrichment in CENH3-ChIPseq of the RepleatExplorer clusters. **b** The monomer sequence of two CL1 satellite variants, Chio1 and Chio2 and their sequence differences are marked (black line). Dyad symmetries are indicated by arrows. The eight nucleotides (purple line) enable formation of hairpin structure between two Chio monomers. **c** Chromosome-level scaffolds of *C. japonica*. Mapping of the centromere repeat Chio1 (red) shows a total of 100 centromere units in the genome assembly of *C. japonica*. **d** The size of centromere units in the 12 *C. japonica* chromosomes. The average centromere size is indicated

as red dots. **e** Chio1 and Chio2 satellite variants intermingle and form mixtures of forward- and reverse-oriented arrays. **f** Hairpin loop structure formed by two Chio1 satellite repeats. **g** Chio1 satellite repeats locate in the knob-like structures (arrows) of pachytene chromosomes. **h** Immuno-FISH shows colocalization of CENH3 (green), Chio1 (purple), and Chio2 (gray) repeats in interphase nucleus and metaphase chromosomes. Chromosomes were counterstained with DAPI. **g**, **h** At least two independent experiments were carried out to confirm the reproducibility of the labeling patterns. **a**, **d** Source data are provided as a Source Data file.

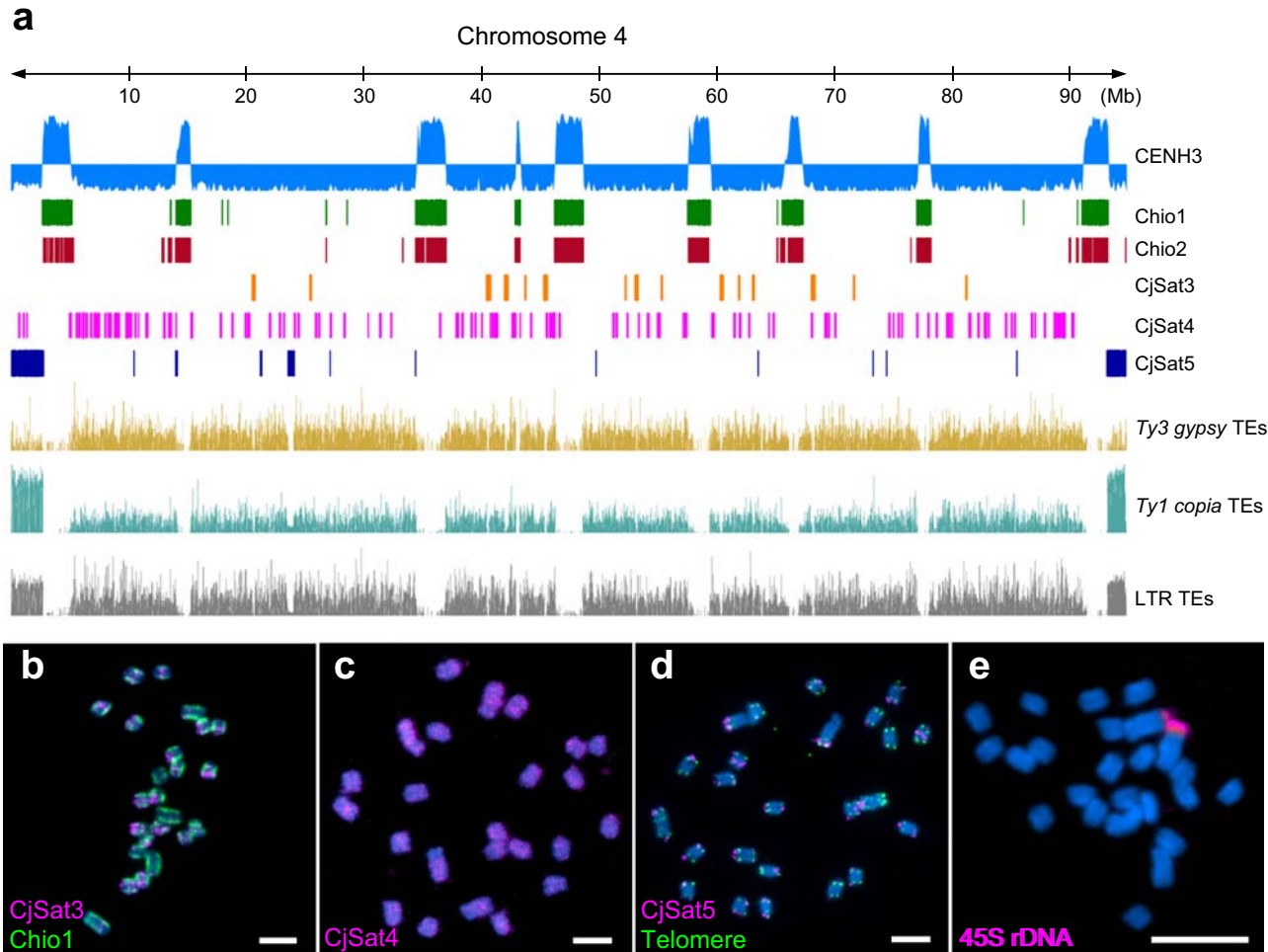

**Fig. 4 | The distribution of CENH3-interacting sequences, high-copy satellite repeats and retrotransposable elements in the genome of *C. japonica*. a** The centromeric Chio1 and Chio2 satellite arrays coincide with the enriched sites of CENH3. The LTR transposable elements (TEs), including *Ty3/gypsy* and *Ty1/copia* TEs, show uniform distribution in the (peri)centromeric and intercentromeric regions. Chromosome 4 is taken as a representative chromosome. **b**–**e** FISH on mitotic *C. japonica* chromosomes with different repeats. **b** CjSat3 shows a clustered distribution, while (**c**) CjSat4 and (**d**) CjSat5 show dispersed and subtelomeric localization, respectively. **e** The 45S rDNA locates at one end of a chromosome pair. Chromosomes were counterstained with DAPI. Scale bar, 5 μm. **b**–**e** At least two independent experiments were carried out to confirm the reproducibility of the labeling patterns.

Contrary to the centromeric regions, intercentromeric regions are transcriptionally active, as revealed by a high correlation between H3K4me2 and counts of mapped transcriptome reads ($r = 0.77$) (Fig. 5c, Supplementary Fig. 12). In general, genes were mainly concentrated in intercentromeric regions (Fig. 5a). The average distance of the closest neighboring gene to centromeric regions was 26.5 Kb.

Gene bodies were highly enriched for mCpG in *C. japonica*, with a sharp decrease at promoters and terminal regions (Fig. 5d). Methylation in the CHH and CHG contexts was lower for the gene bodies than for intergenic regions (Fig. 5d). Remarkably, Chio repeats were highly enriched for mCpG at similar levels to those for TEs (Fig. 5d). mCHG was sharply enriched flanking Chio repeat arrays regions, resembling the H3K9me2 pattern (Fig. 5d). TEs showed the highest enrichment for mCpG and mCHG, while Chio repeats and TEs displayed lower levels of mCHH, similar to genes (Fig. 5d). Our results argue for the presence of a pericentromere-like chromatin state flanking the centromere units in *C. japonica* that may mark the borders for CENH3 loading. Such a similar pattern has been recently reported for the repeat-based holocentromeres in *R. pubera*[17], and is alike to what has been found in monocentric plants like *A. thaliana*[33].

After indirect immunostaining, nuclear chromocenters were H3K4me2-reduced and rich in H3K9me2 (Fig. 6a). At metaphase, anti-H3K9me2 signals mirrored a holocentromere-like labeling pattern at the poleward peripheries of chromosomes, while H3K4me2 is enriched throughout chromosomes except in (peri)centromeric regions (Fig. 6b). Likely, the different condensation degree of eu- and heterochromatin is the reason why intercentromeric and (peri)centromeric regions showed after indirect immunostaining of nuclei and chromosomes an even more contrasting distribution of both types of chromatin compared to the patterns obtained by ChIPseq analysis.

The spatio-temporal pattern of DNA replication after EdU incorporation revealed uniformly labeled metaphase chromosomes when EdU was applied at early S phase (Fig. 6c, Supplementary Fig. 13a–b). After incorporating EdU at mid S phase, the (peri)centromeric regions of chromosomes became stronger labeled and when EdU was applied at late S phase, only the (peri)centromeric chromatin was labeled. Interphase nuclei revealed corresponding patterns (Supplementary Fig. 13a). Thus, the *C. japonica* genome is organized in distinct early and late replicating domains.

## Polymer-based modeling of the holocentromere dynamic in *C. japonica*

We asked how is the formation of a holocentric chromosome structure possible if each chromatid contains only a few megabase-scale

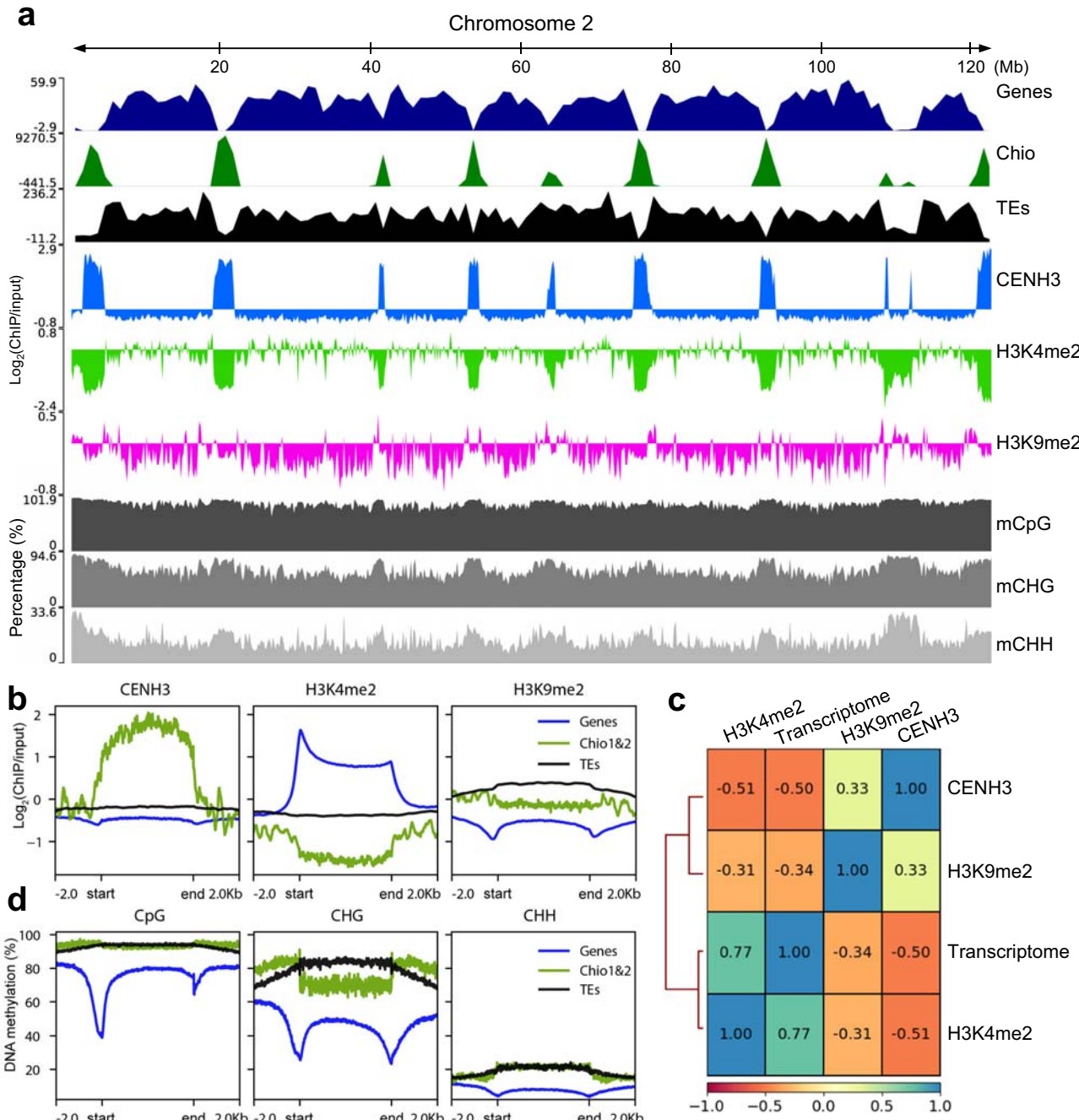

**Fig. 5 | The genome of *C. japonica* is organized in large-scale eu- and heterochromatic regions. a** Distribution of annotated genes, high-copy centromeric Chio satellite repeats, and transposable elements (TEs), as well as the enrichment of CENH3, H3K9me2, and H3K4me2 ChIPseq. The ChIPseq signal tracks are represented as the average log₂ ratio of ChIP/input in genome-wide 10 kb windows. Chromosome 2 is taken as a representative chromosome. DNA methylation level at CpG, CHG, and CHH is shown in percentage. **b** Genome-wide enrichment of CENH3, H3K4me2, and H3K9me2 at different types of sequences, including genes (blue), centromeric Chio satellite arrays (green), and TEs (black). **c** The heatmap shows the correlation scores among different ChIPseq samples (CENH3, H3K9me2 and H3K4me2) and transcriptome. **d** The methylation level of CpG, CHG, and CHH at genes (blue), Chio arrays (green), and TEs (black).

centromere units? In a previously proposed holocentric model[7], the interaction between centromere units and structural maintenance of chromosomes (SMC) proteins is essential during the process of chromosome condensation. Thousands of small centromere units were spread over the genome, resembling the observed distribution of centromere units in *R. pubera*[17]. To form a metaphase holocentromere, short chromatin loops between each centromere unit, which anchored SMC proteins, brought them together into a line[7].

To address the formation of a line-like holocentromere in *C. japonica*, we designed a polymer model based on the distribution of

eight large centromeric units clustered in chromocenters at interphase according to our findings (Fig. 7a). In this model, centromeric nucleosomes attract each other more than non-centromeric nucleosomes and form denser chromocenter-like structures at interphase (Fig. 7b, Supplementary Movie 3). Replicate simulations, which started with random conformations, showed that these chromocenters are sometimes formed by more than one centromeric unit. This is consistent with the lower number of visualized nuclear chromocenters, compared to the number of centromeric units identified in the genome sequence.

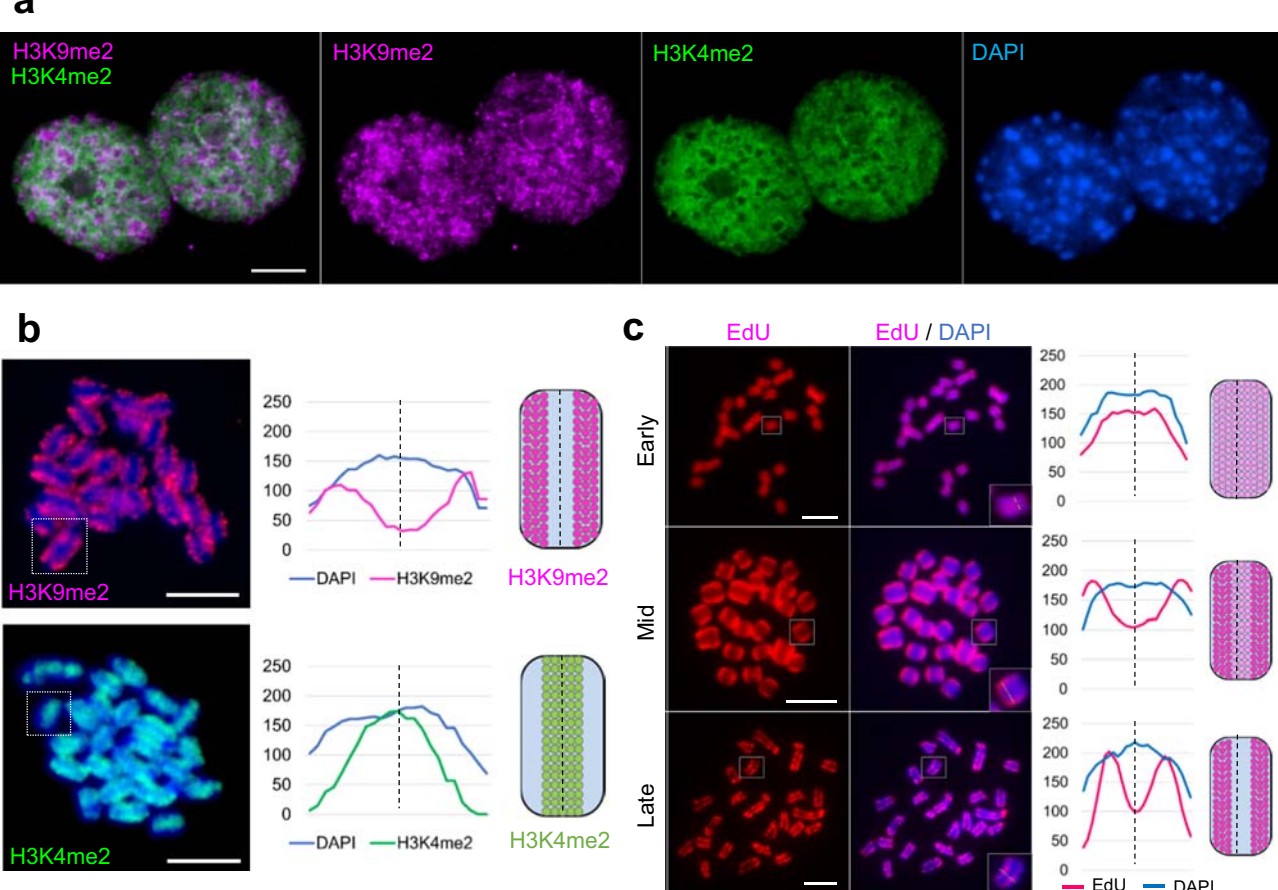

**Fig. 6 | Visualization of eu- and heterochromatic regions of *C. japonica* nuclei and chromosomes.** The immunolabeling patterns of histone H3K9me2 (purple) and H3K4me2 (green) in (**a**) interphase nuclei and (**b**) in metaphase chromosomes confirm the large-scale eu- and heterochromatin organization. **c** EdU labeling patterns (purple) show the DNA compartments replication at early, mid, and late S phases. **b**, **c** The line scan plot profiles show the signal intensities of histone marks,

EdU and DAPI measured in the framed chromosomes (squares). Chromosomes were counterstained with DAPI. Signal distribution along single chromosomes is depicted as schemata next to the profiles. Scale bar, 5 μm. **a**–**c** At least two independent experiments were carried out to confirm the reproducibility of the labeling patterns. **b**, **c** Source data are provided as a Source Data file.

Next, we simulated the condensation of a single *C. japonica* chromatid using the previously proposed loop extrusion mechanism[7]. Loop extruders were prohibited inside the centromeric units, but they were anchored by centromeric nucleosomes at their boundaries (Fig. 7a). Throughout this process, the specific attraction between centromeric nucleosomes persisted, preventing the centromeric units from creating a long line with short chromatin loops in between. Thus, chromatin loops accumulated in the vicinity of the chromocenters, while chromocenters remained condensed by the applied attraction force, resulting in a non-uniformly condensed chromosome (Fig. 7b, c, Supplementary Movie 3). In cytological experiments, we observed a similar structure in prometaphase chromosomes of *C. japonica* (Supplementary Fig. 14a), which is distinct from the smooth prometaphase chromosome of *R. pubera* (Supplementary Fig. 14b). When the chromosome is more condensed, the centromeric units are stretched towards each other forming a line-like holocentromere (Fig. 7b, c), and the differences in density vanish into separate uniformly condensed blocks, one for each region between centromere units (Fig. 7b, c). Thus, our simulation suggests that generally, except for the higher attraction of centromeric nucleosomes, a similar condensation mechanism as modeled for holocentric species possessing numerous small-size chromosome units enables the formation of a holocentromere composed of a few megabase-sized centromere units.

## Discussion

### The interplay between centromere architecture and (epi)genome organization

We report a hitherto unknown type of repeat-based holocentromere organization brought about by strikingly few, evenly spaced megabase-scale CENH3-positive centromere units composed of 23 and 28 bp-long satellite repeats. Also, the fraction of total centromeric DNA in the *C. japonica* genome (16.11%) is exceptionally high, compared to other holocentric species harboring repeat-based centromeres, e.g., <4% of genome DNA is associated with centromeres in *R. pubera*[5,17], and about 3% in the nematode *Meloidogyne incognita*[14].

The centromere units of all other known repeat-based holocentric species are significantly smaller and more abundant. In *M. incognita*, 45–83 bp-long centromeric satellite variants form arrays only up to 1 kb in size[14]. In *R. pubera*, the 172 bp-long Tyba repeat forms 15–25 kb-long (on average 20.5 kb) centromere units, and each chromosome possesses 448–727 units[17]. In contrast, each *C. japonica* chromosome possesses only 7–11 centromere units, whose sizes vary between 0.24 to 4.46 Mb (on average 1.88 Mb). Further, the monomer size of the centromere-associated Chio satellite repeat is below the typical monomer size of 100–400 bp for centromeric repeats[1]. However, centromeric satellites with smaller monomers were also identified in monocentric species[34].

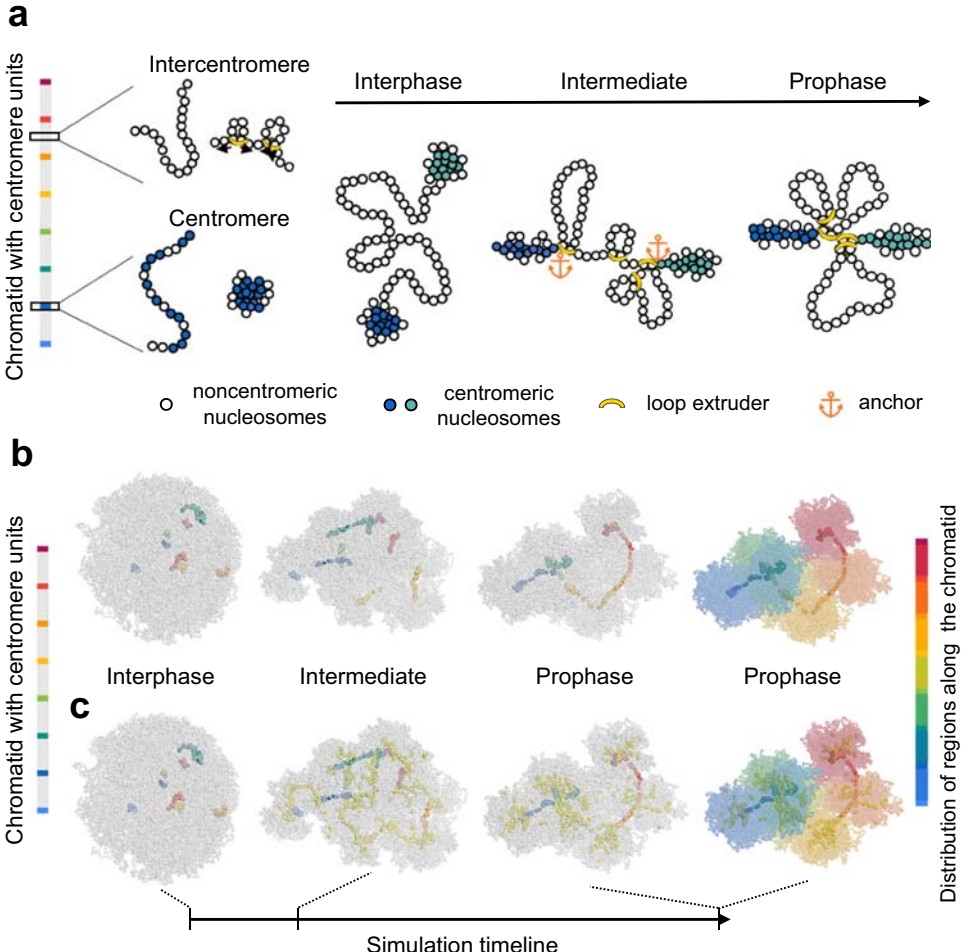

**Fig. 7 | Polymer model and simulated condensation mechanism for a holocentric chromatid based on few and large centromeric units. a** Schema of the proposed condensation mechanism. The color bar on the left indicates the distribution of centromere units along the chromatin fiber of a single chromatid. The chromosomal 10 nm chromatin fiber is represented as a beads-on-a-string polymer. Each centromere unit is made of 60% centromeric nucleosomes (colored beads), which interact more firmly and closely than intercentromeric nucleosomes (non-colored beads). Nucleosomes in intercentromere regions are subjected to loop extrusion by loop extruders (yellow rings), which are anchored by centromeric nucleosomes. The final prophase-like conformations present centromere units condensed by their intrinsic self-attraction and intercentromere regions condensed by loop extrusion. **b** Simulated condensation process of a single holocentric chromatid. The distribution of centromere units is shown in the bar on the left. In the interphase-like conformation, the centromere units are more condensed than the chromatin fiber of intercentromeric regions (gray). An intermediate conformation shows an early step of condensation by loop extrusion. The prophase-like conformation represents a steady state after condensation by loop extrusion. **c** The conformations show the binding of loop extruders (yellow beads) with nucleosomes. The last prophase-like conformation shows the arrangement of colored intercentromere regions, according to the bar on the right. The simulation timeline is indicated as an arrow line.

In monocentric species, megabase-scale centromeric repeat arrays are commonly found. Human centromeres range from 340 kb up to 4.8 Mb (Altemose et al.[35]) and *A. thaliana* centromeres from 2.14 to 2.77 Mb[33]. Thus, the size of single centromere units in holocentric *C. japonica* is comparable to the size of centromeric arrays in monocentric species and is 200-fold larger than those of holocentric *R. pubera*. The average distance between centromeric units on the 12 *C. japonica* chromosomes varies from 7.58 to 11.64 Mb (on average 9.97 Mb), a distance short enough to stably maintain dicentric chromosomes with two active centromeres (~20 Mb in humans[36]). In *C. elegans*, the distance between individual centromere units ranges from 290 bp to 1.9 Mb, with a median of 83 kb[37]. Also, the frequency of centromere units in *C. japonica* is 20 times lower than in *R. pubera*, 0.09 versus 1.88 units/Mb, and the average distance of the closest neighboring gene to centromeric regions was 26.5 kb and 6.3 kb in *C. japonica* and *R. pubera*, respectively. Consequently, in theory, holocentric chromosomes with few centromere units should have after induced DNA double-strand breaks a lower chance of forming centromere-containing chromosomal fragments than those with higher centromere densities.

The large-scale eu- and heterochromatin arrangement of chromosomes and interphase nuclei differs between holocentric species with few large centromere units and those with many small units. While in the latter, eu- and heterochromatin marks are uniformly distributed[17,30], in *C. japonica*, reminiscent of the situation in many monocentric species, centromeres cluster and form chromocenters in interphase nuclei. However, at metaphase, both types of chromatin in holocentrics are arranged side by side from telomere to telomere in a line-like manner. The association of megabase-sized centromeric satellite repeats and the scattered distribution of genic sequences and non-centromeric repeats in intercentromeric regions explain the almost nonoverlapping of both types of chromatin at the chromosomal level in *C. japonica*. However, at sequence level, euchromatic intercentromeric regions possess H3K9me2 sites too, probably due to dispersed and silenced retroelements. The observed DNA replication patterns also confirmed the two defined chromatin states and their corresponding

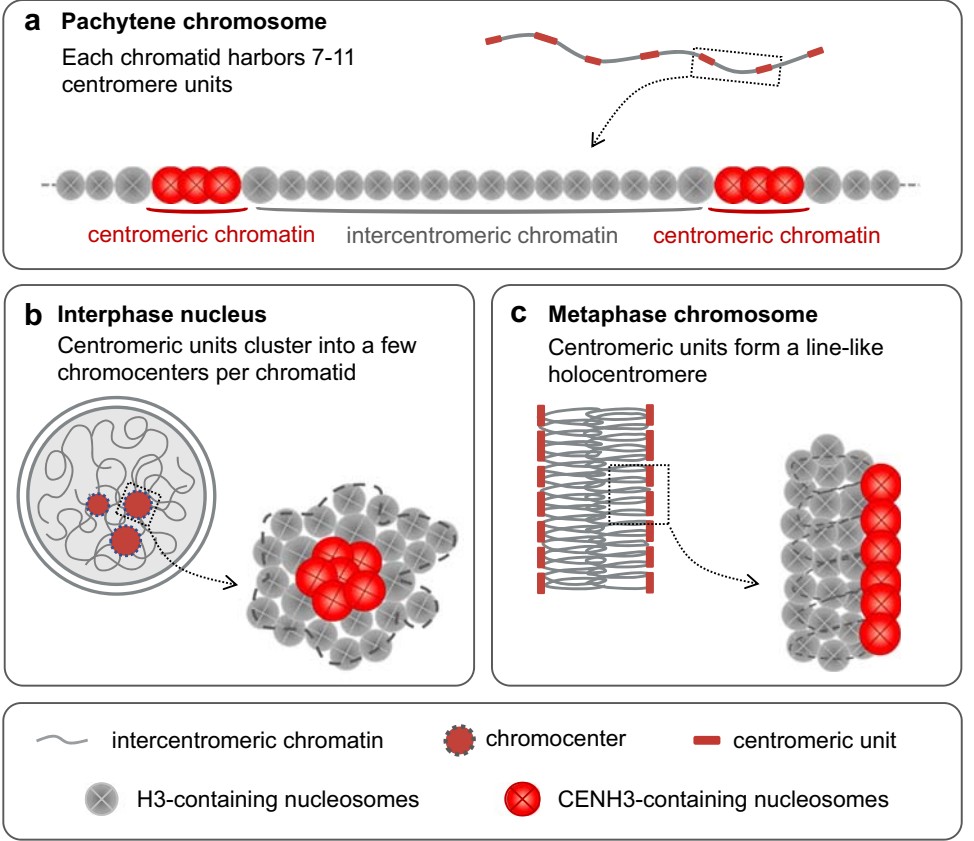

**Fig. 8 | A simplified model of the dynamic organization of centromere units and intercentromeric regions at pachytene, mitotic metaphase and interphase of *C. japonica*. a** Each *C. japonica* chromatid harbors 7–11 evenly spaced megabase-sized centromeric units (red) separated by intercentromeric regions (gray). At pachytene, the chromosome is decondensed and individual centromere units are distinguishable. **b** At interphase, centromeric units cluster into a few chromocenters per chromatid. **c** At metaphase, centromeric units form a line-like holocentromere at the periphery of chromosomes. **a**–**c** Overlapping of CENH3-containing nucleosomes indicates a higher condensation degree of centromeric chromatin.

territories. Thus, in *C. japonica*, despite the monocentromere-like units assembling into a line-like holocentromere at metaphase, the (epi) genome states and fine-scale transcriptional regulation remain unchanged, demonstrating the plasticity of holocentric chromosome organizations.

### The formation of holocentromere in *C. japonica*–a matter of chromosome folding

The evenly spaced centromere units in *C. japonica* might be a prerequisite for the formation of cylindrically shaped metaphase chromosomes with line-like sister holocentromeres facing opposite poles. To assemble the 7–11 megabase-sized centromere units per chromatid into a line-like holocentromere, during mitotic chromosome condensation, looping and folding of chromatin bring the centromere units close to each other to function like a single centromere. Polymer simulations with modulated interaction strengths between centromeric units were used to model the large-scale reorganization of the centromere units during the transition from interphase to mitotic prometaphase, when the clustered interphase chromocenters transformed into line-like holocentromeres. The model is limited in reproducing the interaction between centromeric nucleosomes, where chromocenters appear denser than they really are. We propose that histone H3K9 methylation and/or satellite DNA recognizing proteins mediate "cohesive/sticky" forces between and within chromocenters. However, proteins which mediate satellite DNA clustering in chromocenters are still undiscovered in plants. In animals, the clustering of centromere units at interphase is mediated by proteins bound to pericentromeric satellites (Jagannathan et al.[38]). A summarizing and simplified model of the dynamic organization of centromere units and intercentromeric regions during pachytene, mitotic metaphase and interphase is shown in Fig. 8.

In holocentrics, chromatin folding during chromosome condensation brings distinct centromere units together. At the same time, cooperation between centromeric units and suppression of epigenetic silencing of neighboring centromere units are prerequisites for the evolution and function of a holocentromere. Importantly, we observed that centromeres and microtubules interacted after establishment of line-like holocentromeres and breakdown of the nuclear membrane at prophase. To attract microtubule fibers, individual centromere units join and act as single holocentromere. The arrangement of sister holocentromeres in a back-to-back manner and the close proximity of centromere units at metaphase likely favors the orientation of sister centromeres towards opposite poles. Thus, both spatial arrangement and temporal regulation of centromere units enable the stabilization of the holocentromere in *C. japonica*.

### How did a repeat-based holocentromere evolve?

Understanding the mechanisms that drive rapid expansion, rearrangement, and movement of satellite DNA across the genome is a necessary step in determining the evolution of repeat-based holocentromeres. Different scenarios could explain the transition from mono- to holocentricity. The discovery of metapolycentric chromosomes represents likely a transition from repeat-based mono- to holocentromeres[39]. Metapolycentric chromosomes have centromeric repeat-containing extended primary constrictions which can occupy as much as one-third of the length of a chromosome[11] and might be an

intermediate type of centromere. Is the holocentromere of *C. japonica* a result of a further extended metapolycentromere? A closely related metapolycentric species is unknown. However, the centromeres of the closely related monocentric species *Chamaelirium luteum* are characterized by exceptionally large monocentromeres[40]. Possibly the divergence of the two disjunct distributed genera occurred around 23.5 million years ago and was accompanied by a change of centromere type[41]. The 'macrocentromere' in *C. luteum* might be a precursor to the holocentromere in *C. japonica*[40]. Alternatively, both centromere variants evolved independently.

Centromeric satellite sequence turnover is well established, and differences in copy number and distribution of satellite repeats can be significant between species[42]. Genetic drift is possible, and at least two mechanisms could explain the increase of centromere units along chromosomes and the spread of centromere arrays[43]. Interlocus gene conversion via 3D interaction or multiple inversions with one breakpoint in centromeric satellite arrays during interphase could have facilitated the spreading of the centromeric satellite DNAs. In *C. japonica*, the different size of centromeric repeat arrays, ranging from 0.24 to 4.46 Mb, indicates the dynamic turnover of the centromeric satellite repeats. At interphase, the 7–11 centromere units formed on average only 2.8 chromocenters per chromosome, suggesting associations between about three centromere units per chromosome at interphase, which potentially enable the spreading of the centromeric satellite DNAs via interlocus gene conversion.

Alternatively, a spontaneous burst and spreading of centromeric satellite DNA-containing extrachromosomal circular DNA (eccDNA) and subsequent reintegration into new loci along chromosomes might have occurred. EccDNA accumulation is tightly associated with genome instability and most likely originated from repetitive sequences via erroneous DSB repair[44]. Also, centromeric satellite DNAs were found in the eccDNA fraction in plant species[45].

The Helitron transposable element-mediated dispersal and expansion of holocentromeric Tyba arrays was suggested for *R. pubera*[17]. Such a mechanism is less likely in *C. japonica*. Because first, we found no sequence similarity between Chio repeats and the annotated transposable elements. Second, the size of Chio arrays is on a megabase scale, much larger than the full-length transposable elements of up to 25 kb. Alternatively, the formation of Chio arrays was most likely a step-wise process, with a first seeding of a short Chio array, followed by rounds of expansion through, e.g., microhomology-mediated gene conversion or eccDNA integration into DSBs.

Although the *C. japonica* holocentromere is composed of only a few monocentromere-like units and minor interstitial *Arabidopsis*-type telomere FISH signals were observed, its chromosomes are less likely a product of multiple chromosome fusion events. To achieve a set of 12 chromosomes carrying an average of 8.3 centromere units each, almost one hundred monocentric chromosomal fragments are required. Further, the allied monocentric species *C. luteum* possesses the same chromosome number as *C. japonica*.

Considering the numerous changes and preconditions required to form a holocentromere might help to explain why successful holocentromeres were rarely formed during the evolution of eukaryotes. No case of a return from holo-to-monocentricity has been reported. Thus, once a functional holocentromere is created, it stays and becomes a constitutive feature of the species. The unknown is why holocentromeres evolved by convergent evolution only in some eukaryotic lineages, including invertebrates and plants. The likelihood of forming a holocentromere differs probably depending on the composition, regulation and complexity of the constitutive centromere-associated network (CCAN). However, which component of centromere supports holocentromere formation is unknown, although CENH3 spreading via DNA double strand breacks[46], is a possible candidate. The transitions from mono- to holocentromere are likely based on various evolutionary scenarios rather than on only one

common key event[47]. In summary, our findings broaden the knowledge of the plasticity and diversity of holocentromere organization. We demonstrate the unique value of analyzing non-model species for evolutionary comparison to reveal novelties in even well-studied structures.

## Methods

### Plant materials and in vitro root culture
*Chionographis japonica* plants were grown in a shaded greenhouse: 16 h light (from 6 AM to 10 PM), day temperature 16 °C, night temperature 12 °C. Plants of *Rhynchospora pubera* (Vahl) Boeckler ($2n = 10$) and *Luzula elegans* (Lowe) ($2n = 6$) were cultivated in humid and long-day (13 h light/11 h dark, 20 °C/16 °C) conditions in a greenhouse, and *Arabidopsis thaliana* (Col-0) plants were in the long-day condition of 16 h/8 h, 20 °C/18 °C.

In vitro root cultures of *C. japonica* were induced from leaf petioles. After gentle washing with water, petioles were surface sterilized with a diluted sodium hypochlorite solution (3% active chlorine) supplemented with two drops of Tween 20 for 15 min, followed by a four-times rinse in autoclaved distilled water. Afterwards, petioles were cut into 3 mm segments under sterile conditions and cultivated on ½ Macro Murashige/Skoog (½ MS) medium[48] supplemented with 10.74 μM NAA, 0.44 μM BAP, 3% sucrose, and 0.8% Micro Agar, pH 5.8, in Petri dishes. The parameters in the growth chamber were 16 h light exposure at 26 °C followed by 8 h darkness at 21 °C. To avoid light stress, the explants were shadowed with paper sheets for two weeks. Three weeks later, the petiole segments were transferred to fresh medium. After additional six weeks, roots formed on the petiole segments were separated, multiplied, and subcultured on ½ MS medium supplemented with 2.69 μM NAA, 20% sucrose, 1 g/l peptone, 230 mg/l NaH$_2$PO$_4$ × 2 H$_2$O and 2.5 g/l Phytagel™, pH 5.2, under the same growing conditions. Roots were further subcultured on fresh medium every 4–6 weeks.

### Flow cytometric analysis and flow sorting of G1 nuclei
To isolate nuclei, -0.5 cm$^2$ of fresh leaf tissue of *C. japonica* was chopped together with equivalent amounts of leaf tissue of either of the two internal reference standards *Glycine max* (L.) Merr. convar. max var. max, cultivar 'Cina 5202' (Gatersleben genebank accession number: SOJA 392; 2.21 pg/2C) or *Raphanus sativus* L. convar. sativus, cultivar 'Voran' (Gatersleben genebank accession number: RA 34; 1.11 pg/2C), in a petri dish using the reagent kit 'CyStain PI Absolute P' (Sysmex-Partec, Germany) following the manufacturer's instructions. The resulting nuclei suspension was filtered through a 50-μm CellTrics filter (Sysmex-Partec, Germany) and measured on a CyFlow Space flow cytometer (Sysmex-Partec, Germany) using the FloMax Operating and Analysis Software for Flow Cytometry Particle Analysing Systems Version 2.82 (Supplementary Fig. 15a, b). Ten independent measurements were performed. The absolute DNA content (pg/2C) was calculated based on the values of the G1 peak means and the corresponding genome size (Mbp/1C), according to ref. 49.

For sorting of G1 nuclei, roots were fixed in 4% paraformaldehyde in Tris buffer (10 mM Tris, 10 mM EDTA, 100 mM NaCl, 0.1% Triton X-100, pH7.5) for 5 min on ice under vacuum treatment, followed by another 25 min on ice. After washing twice in ice-cold Tris buffer, the fixed root meristems were chopped in LB01 nuclei isolation buffer[50], filtered as described above and stained with 1.5 μg/ml 4',6-diamidino-2-phenylindoline (DAPI). The nuclear populations were pre-gated in a DNA fluorescence/side scatter plot and the sorting gate for the G1 nuclei was finally defined in a histogram showing the DNA fluorescence. Nuclei were sorted using a BD Influx cell sorter (BD Biosciences, USA) with BD FACS Sortware Version 1.2.0.142 (Supplementary Fig. 15c, d).

### Illumina sequencing of DNA and RNA
Genomic DNA of *C. japonica* was extracted from leaf tissue using the innuPREP Plant DNA kit (Analytik Jena, Germany). Low-pass paired-end

(2 × 150 bp) genome sequencing was performed using Illumina NovaSeq6000 system by Novogene (UK). Total RNAs from leaf, root, flower, and fruit tissues were isolated using the Spectrum™ Plant total RNA kit (Sigma, USA, cat. no. STRN50). Library preparation (Illumina Stranded mRNA Prep Ligation Kit, average library size: 345 bp) and sequencing (paired-end 2 × 151 cycles, Illumina NovaSeq6000 system at IPK Gatersleben) involved standard protocols from the manufacturer (Illumina Inc., USA).

### Repeat analysis
Genomic Illumina PE reads of *C. japonica* were assessed by FastQC[51] available at the RepeatExplorer Galaxy server (https://repeatexplorer-elixir.cerit-sc.cz/galaxy/) and filtered by quality with 95% of bases equal to or above the cut-off value of 10. Qualified PE reads equivalent to 0.5× genome coverage were applied as input to analyze repetitive elements by a graph-based clustering method using RepeatExplorer2 pipeline[28]. The automatic annotation of repeat clusters was manually inspected and revised if necessary, followed by a recalculation of the genome proportion of each repeat type. The genome-wide protein domain-based annotation of transposable elements in *C. japonica* was performed using the REXdb database Viridiplantae v3.0[52], DANTE (https://github.com/kavonrtep/dante) and DANTE-LTR tools (https://github.com/kavonrtep/dante_ltr) implemented in the RepeatExplorer server.

### Transcriptome-based gene identification
The clean RNA-seq datasets from root, leaf and root tissues of *C. japonica* (the umbrella project no. PRJEB58432) were assembled de novo with Trinity 2.4.0[53,54] using default parameters. Putative protein sequences were translated from Trinity contigs that had open reading frames of at least 100 codons. CENH3, MIS12 and NDC80 protein sequences were identified using blastp with homologous protein sequences (XP_038988252.1, XP_008783736.1, and XP_008812729.1, respectively) from *Phoenix dactylifera* (Arecaceae, Liliopsida) as queries.

### Isolation of HMW DNA, HiFi library preparation, and sequencing
For long-read PacBio sequencing, high-molecular weight (HMW) DNA of *C. japonica* was isolated from root cultures using the NucleoBond HMW DNA kit (Macherey Nagel, Germany), quality was assessed with a FEMTOpulse device (Agilent, USA), and quantity was measured by the Quantus fluorometer (Promega, USA). A HiFi library was then prepared according to the "Procedure & Checklist - Preparing HiFi SMRTbell® Libraries using SMRTbell Express Template Prep Kit 2.0" manual with an initial DNA fragmentation by Megaruptor 3 (Diagenode, Belgium) and final library size binning into defined fractions by SageELF (Sage Science, USA). Size distribution was again controlled by FEMTOpulse (Agilent, USA). Polymerase-bound SMRTbell complexes were formed according to standard protocols (Pacific Biosciences of California Inc., USA) and loaded at an on-plate concentration of 85 pM (14, 15, 20, and 26 kb mean length). SMRT sequencing was performed using one 8 M SMRT cell per library (30 h movie time, 2 h pre-extension time) on the Pacific Biosciences Sequel II device, generating a total of 80 Gb (HiFi CCS). The SMRTbell libraries were sequenced at IPK Gatersleben.

### Chromosome conformation capture (Hi-C) sequencing and analysis.
Hi-C sequencing libraries were generated from in vitro root culture of *C. japonica* essentially as described previously[55], and were sequenced (v1.5 chemistry, paired-end, 2 × 111 cycles) using the NovaSeq6000 device (Illumina Inc., USA) at IPK Gatersleben.

### Genome assembly
HiFi reads obtained by the PacBio sequencing were subjected to assembly using the Hifiasm assembler[56] with the command: *hifiasm -o output.asm -t 40 reads.fq.gz*. Preliminary assemblies were evaluated for contiguity and completeness with BUSCO[57] against the Liliopsida_odb10 dataset[58].

### Hi-C scaffolding
Hi-C reads were first mapped to the primary contigs file obtained from the Hifiasm assembler using BWA[59] following the hic-pipeline (https://github.com/esrice/hic-pipeline). Hi-C scaffolding was performed using SALSA2 (https://github.com/marbl/SALSA)[60] with default parameters using 'GATC' as restriction site. After testing several minimum mapping quality values of bam alignments, final scaffolding was performed with MAPQ10. Several rounds of assembly correction guided by Hi-C contact maps and manual curation of scaffolds were performed to obtain the 12 pseudomolecules.

### Antibody production
The synthesized peptides of CENH3 (CjCENH3: MARTKHFSS NRTSRSRKSLRLKQ-C), MIS12 (CjMIS12: C-FAVPEGFVLPKAQDSSG), and NDC80 (CjNDC80: QTVNVRDAERMKRELQAVER-C), were used for immunization of rabbits to generate polyclonal antibodies. The peptide synthesis, immunization, and antibody purification were performed by LifeTein (www.lifetein.com, USA).

### Western blot analysis
To isolate nuclei, young leaves were ground in liquid nitrogen and the powder was collected in lysis buffer (20 mM Tris-HCl pH 7.4, 25% glycerol, 20 mM KCl, 2 mM EDTA, 2.5 mM MgCl$_2$, 250 mM sucrose). To overcome the viscosity of the leave extract resulting from high polysaccharide content, the nuclei purification was carried out in 10× volume of lysis buffer. Concentration of nuclear proteins was determined using the Bradford assay (Protein Assay Kit II, Bio-Rad, USA). Nuclear protein extract (20 μg) was loaded onto a 10% SDS-PAGE gel and separated at 100 V for 2 h using a Mini Protean® Tetra Cell system (Bio-Rad, USA). Proteins were electro-transferred onto Immobilon TM PVDF membranes (Millipore, USA). The membrane was incubated with rabbit anti-*C. japonica* CENH3 antibody (dilution 1:1000) at 4 °C for 12 h, followed by a detection with the secondary antibody anti-rabbit IRDye 800CW (dilution 1:5000, LI-COR, USA) in 1× PBS containing 5% w/v low-fat milk powder at 22°C for 1 h. Image was captured using Odyssey (Li-Cor, USA) as recommended by the manufacturer.

### Indirect immunodetection
Mitotic chromosomes and interphase nuclei were prepared from root meristems fixed in 4% paraformaldehyde in Tris buffer (10 mM Tris, 10 mM EDTA, 100 mM NaCl, 0.1% Triton X-100, pH7.5) for 5 min on ice under vacuum treatment, followed by another 25 min solely on ice. Root meristems were then chopped in lysis buffer LB01 (15 mM Tris, 2 mM Na$_2$EDTA, 0.5 mM spermine, 80 mM KCl, 20 mM NaCl, 15 mM β-mercaptoethanol, and 0.1% Triton X-100)[50], the cell suspension was filtered through a 50-μm CellTrics filter and subsequently centrifuged onto slides using a Cytospin3 (Shandon, Germany) at 700 rpm (×55.32 g) for 5 min. The chromosome spreads were blocked in 5% BSA in 1 × PBS at room temperature (RT) for 1 h and incubated with primary antibodies in 1× PBS containing 1% BSA at 4 °C overnight. The slides were washed in 1× PBS at RT for 5 min, three times, and then secondary antibodies were applied, followed by incubation at 37 °C for 1 h. After three washes in 1× PBS at RT for 5 min, the slides were dehydrated in 70-90-100% ethanol series for 3 min each and counterstained with 10 μg/ml DAPI in Vectashield antifade medium (Vector Laboratories, USA). For immunodetection of microtubules, the Tris buffer or 1× PBS mentioned above was substituted by 1× MTSB buffer (50 mM PIPES, 5 mM MgSO$_4$, and 5 mM EGTA, pH 7.2).

The primary antibodies used in this study included customized rabbit anti-*C. japonica* CENH3 (anti-CjCENH3, dilution 1:1000), rabbit

anti-*C. japonica* MIS12 (anti-CjMIS12, dilution 1:100), and rabbit anti-*C. japonica* NDC80 (anti-CjNDC80, dilution 1:100), as well as commercially available mouse anti-alpha-tubulin (Sigma-Aldrich, USA, cat. no. T9026-2, dilution 1:300), rabbit anti-histone H3K4me2 (abcam, UK, cat. no. ab7766, dilution 1:300), mouse anti-histone H3K9me2 (abcam, UK, cat. no. ab1220, dilution 1:200), mouse anti-histone H3S10ph (abcam, UK, cat. no. ab14955, dilution 1:1000), rat anti-histone H3S28ph (Sigma-Aldrich, USA, cat. no. H9908, dilution 1:1000), rabbit anti-H3T3ph (Sigma-Aldrich, USA, cat. no. 07-424, dilution 1:1000), and rabbit anti-H2AT120ph (Active Motif, USA, cat. no. 61196, dilution 1:500).

The anti-rabbit rhodamine (Jack ImmunoResearch, USA, cat. no. 111295-144, dilution 1:400), anti-rabbit Alexa488 (Jack ImmunoResearch, USA, cat. no. 711-545-152, dilution 1:400), anti-mouse Alexa488 (Jack ImmunoResearch, USA, cat. no. 715-546-151, dilution 1:400), and anti-rat Alexa488 (Jack ImmunoResearch, USA, cat. no. 112-545-167, dilution 1:400) were used as secondary antibodies.

### Transmission electron microscopy (TEM)
For electron microscopy analysis, root tips and leaf cuttings, were used for aldehyde fixation, dehydration and resin embedding (see Supplementary Table 5). Ultra-thin sections (70 nm) were cut with a Leica microtome Ultracut S (Leica Microsystems, Wetzlar, Germany), and mounted on 70 mesh cupper TEM grids. Prior to ultrastructure analysis at 120 kV in a Tecnai Sphera G$^2$ transmission electron microscope (ThermoFisher Scientific, Eindhoven, Netherlands), sections were contrasted in a Leica EM AC 20 automatic contrasting device with homemade 2% uranyl acetate for 30 min, followed by a 90-second incubation in Leica Ultrastain 2 containing 3% Reynolds' lead citrate.

### Preparation of labeled fluorescence in situ hybridization (FISH) probes
The consensus sequences of putative satellites reconstructed by TAREAN (TAndem REpeat ANalyzer)[61] were used to design oligonucleotides or primers for probe DNA amplification (Supplementary Table 4). The 5′ FAM and 5′ TAMRA fluorochrome-conjugated oligos and PCR primers were synthesized by Eurofins (Germany). Probe DNAs were amplified in a mixture of 50 ng genomic DNA, 1× PCR buffer, 0.25 mM of each dNTP, 0.4 μM of each primer, 1.5 U Taq polymerase (QIAGEN, Germany), in a total of 50 μl with a program of 95 °C for 5 min, 35 cycles of 95 °C for 30 s, 55 °C for 40 s and 72 °C for 40 s, followed by 72 °C for 5 min. The clones pAt T4[62] and pTa71[63] were used as probes to detect *Arabidopsis*-type telomeres and 45S rDNA loci, respectively. Purified PCR products and plasmid DNAs were labeled with ATTO488-dUTP or ATTO550-dUTP using Fluorescent Nick Translation Labeling kits (Jena Bioscience, Germany).

### Chromosome preparation and fluorescence in situ hybridization (FISH)
Mitotic chromosome spreads of *C. japonica* were prepared from root meristems using a dropping method. Roots were pretreated in ice-cold water for 20–24 h, fixed in 3:1 (ethanol: glacial acetic acid) fixative at RT, overnight and kept in 70% ethanol at −20 °C until use. Fixed roots were digested in an enzyme mixture (0.7% cellulase Onozuka R10 (Duchefa Biochemie, The Netherlands, cat. no. C8001), 0.7% cellulase (Calbiochem, USA, cat. no. 219466), and 1.0% pectolyase (Sigma, USA, cat. no. 45-P3026)) in citric buffer (0.01 M sodium citrate dihydrate and 0.01 M citric acid) at 37 °C for 30–40 min. Cell suspension in the 3:1 fixative was dropped onto slides on a hot plate at 55 °C, and slides were further fixed in 3:1 fixative for 1 min, air-dried, and kept at 4 °C for later use.

To prepare pachytene chromosomes, inflorescences of *C. japonica* in the length of 0.7–1.0 cm were collected and fixed as described above for roots. Anthers were digested at 37 °C for 70–80 min in the enzyme mixture (0.23% cellulase Onozuka R10 (Duchefa Biochemie, The Netherlands, cat. no. C8001), 0.23% cellulase (Calbiochem, USA, cat. no. 219466), 0.33% pectolyase (Sigma, USA, cat. no. 45-P3026), and 0.33% cytohelicase (Sigma, USA, cat. no. C8247)). Meiotic spreads were prepared by a dropping method[64]. Standard FISH was performed with mitotic and meiotic chromosomes.

### Analysis of DNA replication by EdU labeling
DNA replication patterns were visualized by incorporation of 5′-ethynyl-2′-deoxyuridine (EdU) into the newly synthesized DNA strand using the EdU Cell Proliferation Kit (BaseClick, Germany). Roots of *C. japonica* were incubated in a 15 μM EdU-containing Hoagland solution (Sigma, USA) for 2 h at RT. They were then transferred to Hoagland solution and incubated for either 3, 6, 12 or 24 h, followed by fixation in ethanol: glacial acetic acid (3:1) at RT, overnight. Chromosome spreads were prepared by dropping method as described above, and EdU was visualized by click reaction following the kit protocol. The slides were counterstained with 10 μg/ml DAPI in Vectashield antifade medium (Vector Laboratories, USA).

### Microscopy and image analysis
Widefield fluorescence images were captured using an epi-fluorescence microscope BX61 (Olympus Europa SE &Co. KG, Germany) equipped with an Orca ER CCD camera (Hamamatsu, Japan) and pseudo-colored by the Adobe Photoshop 6.0 software. To analyze chromatin ultrastructures, we applied super-resolution spatial structured illumination microscopy (3D-SIM) using a 63×/1.40 Oil Plan-Apochromat objective of an Elyra PS.1 microscope system (Carl Zeiss GmbH, Germany). Image stacks were captured separately for each fluorochrome using the 561, 488, and 405 nm laser lines for excitation and appropriate emission filters. Maximum intensity projections from image stacks were calculated via the Zeiss ZENBlack software (Carl Zeiss GmbH, Germany). Zoom-in sections were presented as single slices to indicate the subnuclear chromatin structures at the super-resolution level. 3D rendering to produce spatial animations was done based on SIM image stacks using the Imaris 9.6 (Bitplane, UK) software. The volumes of CENH3 signals and DAPI-labeled whole G1 nuclei were generated and measured via the Imaris tool 'Surface', and the number of signals was counted. The percentage of colocalized immunolabeling and FISH signals were calculated via the Imaris tool 'Coloc' and the number of signals was detected using the Imaris tool 'Spots'.

### Chromatin immunoprecipitation (ChIP) sequencing
For ChIP, 0.65 g of *C. japonica* flower and 1.0 g of *Secale cereale* (inbred line Lo7) leaf tissue were ground separately with liquid nitrogen and homogenized in 10 ml nuclei isolation buffer (1 M sucrose, 5 mM KCl, 5 mM MgCl$_2$, 60 mM HEPES pH 8.0, 5 mM EDTA, 0.6% Triton X-100, 0.4 mM PMSF, 1 μM pepstatin A, cOmplete protease inhibitor cocktail (Roche, Swiss)) to isolate nuclei. Nuclei fixation was performed in 1% PFA/nuclei isolation buffer at RT, 12 rpm for 10 min and terminated by adding 2 M glycine to a final concentration of 130 mM. The nuclei suspension was filtered through Miracloth (Millipore, USA) twice and a 50 μm Cell-Trics filter (Sysmex-Partec, Germany) once and centrifuged at 4 °C, 3000 × g for 10 min. The nuclei pellet was resuspended in 1 ml extraction buffer (0.25 M sucrose, 10 mM Tris-HCl pH 8.0, 10 mM MgCl$_2$, 1% Triton X-100, 1 mM EDTA, 5 mM β-mercaptoethanol, 0.1 mM PMSF, 1 μM pepstatin A, cOmplete protease inhibitor cocktail), followed by centrifugation at 4 °C, 12,000 × g for 10 min. After removing the supernatant, nuclei were resuspended in 150 μl of nuclei lysis buffer (20 mM Tris-HCl pH 8.0, 10 mM EDTA, 1% SDS, 0.1 mM PMSF, 1 μM pepstatin A, cOmplete protease inhibitor cocktail). Chromatin was sonicated using 14 cycles of 30 s ON, 30 s OFF at high power, in a Bioruptor (Diagenode, USA), followed by adding 100 μl ChIP dilution buffer (16.7 mM Tris-HCl pH 8.0, 167 mM NaCl, 1.1% Triton X-100, 1 mM EDTA, cOmplete protease inhibitor cocktail), and continued sonication to a total of 31 cycles under the same

setting. The sonicated samples were diluted 10 times with ChIP dilution buffer, centrifuged at 4 °C, 13,000 × $g$ for 5 min, and the supernatant of each sample was transferred to new tubes. To dilute the high proportion of the putative *C. japonica* centromeric repeat, sonicated chromatin of *S. cereal* was added to the sonicated chromatin of *C. japonica* in an 8:1 ratio. The mixed chromatin samples were incubated with the CENH3 antibody (10 mg/ml) to a final 1:500 dilution at 4 °C by shaking at 14 rpm for 12 h. Dynabeads™ Protein A (Invitrogen, USA) in ChIP dilution buffer, corresponding to 0.1× volume of the chromatin solution, was added to the antibody-prebound chromatins and incubated at 4 °C by shaking at 14 rpm for 1.5 h. The collected beads were then washed twice in low salt buffer (150 mM NaCl, 0.1% SDS, 1% Triton X-100, 2 mM EDTA. 20 mM Tris-HCl pH 8.0), followed by two washes in high salt buffer (500 mM NaCl, 0.1% SDS, 1% Triton X-100, 2 mM EDTA. 20 mM Tris-HCl pH 8.0), and another two washes in TE buffer at 4 °C by shaking at 14 rpm for 5 min. The bead-bound chromatin was purified by using iPure kit v2 (Diagenode, USA) following the manual and quantified using Qubit™ dsDNA HS Assay kit (Invitrogen, USA). ChIPseq libraries were prepared by NEBNEXT® Ultra™ II DNA Library Prep Kit for Illumina (New England Biolabs, USA) and sequenced using NovaSeq 6000 system (Illumina, USA) by Novogene (UK) in the format of 150 × 2 paired-end reads.

### Methyl-seq analysis
To evaluate DNA methylation, we applied enzymatic Methyl-seq and used the Bismarck pipeline[65] to analyze the data. Individual methylation context files for CpG, CHG, and CHH were converted to BIGWIG format and used as input track for visualization of genome-wide DNA methylation with pyGenomeTracks[66].

### ChIPseq analysis and metaplots
To evaluate the enrichment of repeats associated with CENH3-containing nucleosomes, the single-end reads of CENH3-ChIPseq and input were quality filtered using the tool 'Processing of FASTQ reads', implanted in the Galaxy-based RepeatExplorer (https://repeatexplorer-elixir.cerit-sc.cz/galaxy/) portal. ChIP-Seq Mapper (Galaxy version 0.1.1)[28] was used to map the ChIP- and input reads on RepeatExplorer-derived contig sequences of repeat clusters.

The paired-end illumina reads of ChIPseq (2 × 150 bp) were mapped to the *C. japonica* genome assembly using Bowtie2[67] with default parameters. The multimapped reads were filtered out from the Bowtie2 outputs using Sambamba with options "-F [XS] == null and not unmapped and not duplicate"[68]. CENH3 domains were identified by comparing the ChIP and input data using MACS3[69]. The parameters for MACS3 included -B –broad –g 1380000000 –trackline. As an alternative method for detection of CENH3 domains, we compared input and ChIP using the epic2 program for detection of diffuse domains[70]. Parameters for epic2 included --bin-size 2000. Only CENH3 domains detected with both methods were kept for further analysis. To determine the sizes and positions of centromere units, we merged with bedtools CENH3 peaks that were separated by <500 kb to eliminate the gaps that arise because of fragmented Chio repeat arrays or due to insertion of TEs. Small CENH3 domains of <1 kb were discarded. Length and distance between Chio arrays and between CENH3 domains were then calculated using bedtools.

The deeptools bamCompare[71] was used to generate normalized ChIPseq signal tracks of the average of log2-ratio of read counts in ChIP over input. The generated normalized BIGWIG files were used to calculate the level of enrichment associated with gene bodies, Chio repeats, and TEs using computeMatrix scale-regions (parameters: --region Body Length 4000 –before Region Start Length 2000 –after Region Start Length 2000). Finally, metaplots for all ChIPseq treatment files were plotted with plotHeatmap available from deeptools[71]. In addition, coverage BIGWIG files of transcriptional activity (RNAseq) and all DNA methylation contexts were also used to calculate their enrichment on gene bodies, Chio repeats, and TEs with computeMatrix and plotting with plotHeatmap. The deeptools multiBigWigSummary and plotCorrelation[71] were used to calculate and plot the Spearman correlation between different ChIPseq and RNA-seq targets as a heatmap. Plots of detailed chromosome regions showing multiple tracks were done with pyGenomeTracks[66].

### Polymer simulation
We modeled the chromatin as a polymer chain with 100,000 monomers, each monomer corresponding to one nucleosome with ~200 bp DNA, considering linker regions. The entire polymer corresponds to a theoretical ~20 Mb chromatid, a lower-scale model of an average 85 Mb-long chromatid of *C. japonica*. In this model, centromeric nucleosomes are uniformly distributed inside eight different centromeric units (Fig. 7). Each unit is 442 kb long, with 1325 centromeric nucleosomes (60% of the nucleosomes inside a centromeric unit). Pairs of centromeric nucleosomes have a different attractive-repulsive force between each other, mimicking a selective sticky force. We included 500 loop extruders mimicking the presence of SMC proteins. They were prohibited inside the centromeric units, but dynamically extruded loops outside them, and they were anchored by centromeric nucleosomes at their borders.

We performed Langevin dynamics simulations with OpenMM Python API (Application Programming Interface)[72] as in ref. 7. We applied only three internal forces to obtain a chromatin-like motion: (1) a harmonic force between pairs of consecutive nucleosomes; (2) a bending force between triplets of consecutive nucleosomes; and (3) an attractive-repulsive force between non-consecutive nucleosomes that allows for eventual crossing of the chromatin fiber mimick the presence of topoisomerase II. An attractive-repulsive potential was specially designed for pairs of centromeric nucleosomes ($U_{cc}$) (Eq. 1).

$$U_{cc}(x) = \begin{cases} 5 + 6\left(\left(\frac{x}{8}\right)^4 - 2\left(\frac{x}{8}\right)^2\right), 0 \le x \le 8 \\ 5 + 6\left(\left(\frac{x}{20} + 0.6\right)^4 - 2\left(\frac{x}{20} + 0.6\right)^2\right), x > 8 \end{cases} \quad (1)$$

Compared to the potential between other pairs of nucleosomes, it has a global minimum at $x = 8$ (being x the distance between the two nucleosomes in nm) to ensure centromeric nucleosomes are closer to each other, and it has a slower increase for $x > 8$ to attract nucleosomes at larger distances.

The initial interphase-like conformation was reached after 5,000,000 simulation steps without loop extrusion and confined to a sphere proportional to the average volume of root nuclei in G1 of *C. japonica* (110.57 μm³). After this, simulation of the mitotic condensation process continued with the same potentials, except the spherical confinement, and considering loop extrusion. The loop extrusion simulation was performed first in one dimension as in[7], and then added to the three-dimensional polymer simulation. All simulations lasted 25,000,000 steps. Images of the chromosome model at different condensation steps were made with PyMOL (https://pymol.org/2/)[73].

### Reporting summary
Further information on research design is available in the Nature Portfolio Reporting Summary linked to this article.

## Data availability
The data that support this study are available from the corresponding authors upon reasonable request. The datasets generated for this study can be found in the European Nucleotide Archive (ENA) at EMBL-EBI under the umbrella project no. PRJEB58432. The final processed ChIP-seq datasets were deposited in the NCBI GEO database under the accession number of GSE228407. The REXdb database Viridiplantae v3.0 [http://repeatexplorer.org/?page_id=918] is publicly available. Source data are provided with this paper.

## Code availability

All the Fortran and Python scripts, for running the simulations and for visual analysis in PyMOL are available in BitBucket (bitbucket.org/ipkdg/chionographis_simulations.git).

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

## Acknowledgements

We thank Katrin Kumke (IPK, Germany) for cytological support, Ines Walde, Manuela Knauft and Susanne König (IPK) for expert technical assistance in DNA and RNA sequencing, Marion Benecke and Kirsten Hoffie (IPK) in electron microscopy, Anne Fiebig and Maren Maruschewski (IPK) for sequence submission, Ingo Schubert (IPK) for critical discussion of the manuscript, and Noriyuki Tanaka (Japan) for valuable comments. This work was supported by the Deutsche Forschungsgemeinschaft DFG grants HO1779/32-1 and HO 1779/32-2 to A.H., SO 2132/1-1 to A.S.C., and MA9363/3-1 to A.M.; the Taiwan Ministry of Science and Technology grants MOST 106-2313-B-002-034-MY3 and MOST 108-2811-B-002-608 to Y.T.K.; the Czech Science Foundation grant 20-25440S to P.N.; and the Max Planck Society to A.M. Deutsche Forschungsgemeinschaft (DFG, German Research Foundation, grant 491250510).

## Author contributions

Y.T.K. performed majority of the experiments, including immunostaining, FISH, ChIPseq, repeat identification, DNA replication assay, and data analysis; A.S.C. performed polymer simulation; V.S. performed super-resolution microscopy and image analysis; P.N. and J.Y.C., performed transcriptome analysis; JM performed transposon annotation; M.Me performed electron microscopy; J.F. conducted flow cytometry; S.A., E.K., and F.D. performed tissue culture; B.H. performed PacBio library preparation; A.Hi performed sequencing, D.D. performed Western blot; A.M. and M.Ma performed genome assembly; A.M. performed Hi-C scaffolding, gene annotation and sequence analysis; T.I. arranged plant materials; A.Ho supervised the research project; Y.T.K. and A.Ho wrote the manuscript with the help of all coauthors.

## Funding

## Competing interests

The authors declare no competing interests.
