## [Peer Review File · Nature Communications]

Holocentromeres can consist of merely a few megabase-sized satellite arraysREVIEWER COMMENTS

Reviewer #1 (Remarks to the Author):

In this study the authors provide insights into the genome and nuclear organization of the holocentric plant *Chionographis japonica* using genomic and microscopic approaches.

Using antibodies that they generated against centromere and kinetochore components of *C. japonica* the authors confirmed the presence of holocentric chromosomes in this species with centromeres located over the entire length of mitotic chromosomes. They also found that, similar to some monocentric organisms, *C. japonica* centromeres in interphase overlap with a few large chromocenters. This is unlike other known holocentric organisms where centromere clusters have not been described. Using genomic approaches targeting CENH3 and other histone/DNA modifications they localized CENH3 to a few (~8) megabase-sized arrays per chromosome that are depleted in H3K4me2, flanked by H3K9me2 regions and enriched in CpG methylation. Again, these features resemble the organization of individual centromeres of the monocentric plant *A. thaliana* as well as the holocentric *R. pubera*. Using a bottom-up approach clustering the CENH3-enriched reads followed by DNA FISH experiments as a confirmation they determined that centromeres are enriched for two types of related satellites, *Chio1* and *Chio2*. Interestingly, on mitotic chromosomes staining for the H3K9me2 mark (and late-replicating regions) resembles centromere staining. Finally, using polymer simulation approaches the authors propose a model underlying the structural dynamics of megabase-sized centromeric units clustered into chromocenters in interphase to the formation of a "centromere line" in mitotic chromosomes.

The study is technically sound, the writing is clear and the conclusions follow clearly from the results. It is also interesting to note that holocentromeres can consist of only a few large centromere units, each of which resembles a single monocentromere in other plants. I only have two minor comments.

Figure 2: The CENH3 and MIS12 immunosignals do not form the characteristic lines along the chromosomal surface as in Figure 1 and S3c? It seems to be rather telophase than anaphase as indicated?

Figure 6: The resemblance between the H3K9me2 and centromere immunosignals are interesting. How do the authors reconcile the fact that H3K9me2 is rather enriched in pericentric regions than at the centromere core?

Reviewer #2 (Remarks to the Author):

The manuscript by Kuo et al. describes the unique holocentromeres of *Chionographis japonica*, which consist of ~9 distinct megabase-sized arrays of 23 and 28 bp tandem repeats per chromosome that assemble into a linear kinetochore. The authors also assemble much of the *C. japonicus* genome and map cenH3, H3K4 me2 and H3K9me2 nucleosomes and DNA methylation to address the overall genome architecture. They modify an earlier model of holocentromere condensation of many small centromeric units to account for how a small number of large centromere units can assemble into a linear kinetochore. This paper will be of interest to the fields of centromere biology and chromosomal structure not only because it is another unique example of independently derived holocentromeres but because it helps illuminate how a chromosome might evolve from a monocentromere to a metapolycentric centromere to a holocentromere. The writing is mostly clear, but I have a few mostly minor comments.

On page 6, 2nd paragraph the authors discuss the number of chromocenters and conclude there are 2.8 per chromosome. Although it is easy to do the math, they don't mention here or in Table S1 how

many chromosomes *C. japonica* has until the next page, whereas it would be useful to mention it here. Is the chromosome number known from some earlier cytology (presumably the Tanaka references), or was it deduced from the genome assembly? The former seems more likely, since *C. japonica* was already known to be holocentric. A reference to the earlier work would be appropriate. Although they address the clustering of centromere units into chromocenters in the later modeling section and discussion, it would be helpful to point it out on page 8 where they determine that number of centromere units per chromosome is ~ 9 , not 2.8.

At the bottom of page 6-top of page 7, the authors did not detect H2AT120ph, and present in Fig S6 three H2A sequences, two with alanine in place of threonine at position 120, which cannot be phosphorylated, and one with serine, which possibly could be phosphorylated. Do the authors have information on the transcript abundance of these three genes or any other H2A genes in dividing (or non-dividing) tissues that might support the lack of a phosphorylatable H2A? This is not essential, but would be interesting if the information is available.

On page 8 line 25, the authors specify nine CENH3 regions, but omit to say that this is per chromosome.

In Figure 3e, the authors show that there are a nearly equal number of interspersed 'forward' and 'reverse' blocks of Chio1/Chio2 repeats. This seems unusual, and suggests considerable rearrangement of tandem sequences. The authors mention in the discussion (p. 17) the bridge-breakage-fusion model that might account for inverted sequences, but the authors might choose to elaborate a little on how they imagine this accounting for the distribution of forward and reverse blocks of satellites.

In the legend to Figure 3g (p. 33), the authors refer to arrows but I cannot see any arrows. Please correct the legend or the figure.

In Figure 5b and 5d the x-axis is marked with TSS and TES, which makes sense for genes (blue) and possibly for the transposons (black) but not so much for Chio 1&2 (green). Perhaps these points can be labeled 'start' and 'end' instead of TSS and TES, or there can be an explanation in the legend.

In Figure S12, the legend says Patterns 1-3 correspond to early, mid and late replication, respectively, but in Figure S12b the 'late' pattern 3 shows up at three hours while the 'early' pattern barely shows up at 12 hours and isn't abundant until 24 hours. It appears that the colors for patterns 1 and 3 may have been switched in the inset legend in 12b, or else I don't understand what is being shown. Please correct or clarify.

In movie #2, there appear to be ~ 6 green blobs that vary in size but seem large for telomeres and many fewer than I would expect for 2 ends x 24 chromosomes. These also do not look like clustered telomeres where several chromosomes come together, but rather like ~ 6 grooveless green chromosomes. Also the purple Chio1 appears to be on several chromosomes, but not on all, and does not come together in a line, though the legend says this is metaphase. Please explain what is being shown. The legend merely says that telomeric DNA is green and Chio1 is purple.

The file for movie #3 is corrupt and I could not open it.

Reviewer #3 (Remarks to the Author):

Review for Kuo et al., "Plasticity in centromere organization: Holocentromeres can consist of merely a few megabase-sized satellite arrays".

The Houben lab specialise in investigating chromosome structure and function in plants and are particularly interested in centromere organisation, regulation, and evolution. Their recent Cell paper focused on holocentromeres. Unlike most species that have a single monocentromere visualised as a primary constriction in mitosis, holocentric species form centromeres (100s) along the entire chromosome length (Hoffstatter et al 2022). This current manuscript (Kuo et al) further explores holocentromeres by focusing on another holocentric plant, the lilioid *Chionographis japonica*. Kuo et al assembled the reference genome for *C. japonica* and describe its centromere and epi-genome organisation. This study is specifically interesting as *C. japonica* has only a few (7-11) evenly-spaced megabase-sized centromeres which are CENH3 positive, far fewer and much larger than all other holocentric species described thus far. Interestingly in many of their analyses these holocentromeres function more like monocentromeric than holocentric species with 100s of centromeres along the chromosome, for example *C. japonica* form centromere clusters in chromocenters at interphase and the *C. japonica* genome is organised into distinct eu- and heterochromatin domains, similar to monocentric species. However, at the same time in *C. japonica*, like other holocentromeric species, microtubule attachment and kinetochore proteins colocalise with CENH3 positive regions along the length of the chromosome. These data very nicely demonstrate the plasticity of holocentric chromosome organisation, and the authors provide a great discussion on holocentromere evolution and where *C. japonica* might fit within this. Overall, this is a nicely conducted and compelling study of a novel holocentromere which will broaden our mainly monocentric-based understanding of centromere identification and function.

I have some minor points for the authors to address that should clarify a few questions that came from reviewing this manuscript:

1: The authors have previously shown that in other holocentric species *L. elegans* and *R. pubera* the CENH3- positive centromere forms a longitudinal groove at metaphase. One specifically interesting result with *C. japonica* is that this groove structure is absent.

Do the authors think the absence of a central groove is significant? Could this be related to the absence of H2AT120ph in *C. japonica*?

2: The authors use Supplementary movie 2 to support their finding of a lack of longitudinal groove. I don't find this movie particularly helpful in presenting this. Perhaps a surface rendering of SIM image stacks of one/a few chromosomes (as they have presented previously in Fig.1 Marques et al 2015) would more clearly display the lack of grooves.

It is also not clear to me in this movie why the purple signal of the Chio 1 satellite is not present on all chromosomes as Figure 3C and 3H indicate Chio 1 repeats are found on all chromosomes.

3: I am particularly interested in how the authors have mapped CENH3 ChIP seq reads to specific Chio 1 and Chio 2 repeat regions in *C. japonica*. Can the authors provide more explanation as to how these repetitive short read sequences were mapped to specific regions on each of the 12 different *C. Japonica* chromosomes. My understanding is that as these are repetitive reads, one cannot be certain the specific location of each individual read, therefore these must have been randomly distributed.

Similarly, with the H3K9me2 reads that map to repetitive Chio 1 or 2 sequence, it is not clear to me how the authors have been able to assign these to specific repeat regions. The authors state that "part of the centromere units were depleted of H3K9me2" but I am unclear as to how they have been able to know which exact repeat region these reads were derived from.

4: The authors have previously used polymer-based modelling to propose a mechanism for how holocentric chromosomes form a line-like centromeric structure in metaphase (Camara et al 2021). In this manuscript they use the same approach to describe how, with only a few centromere units along

C. Japonica chromosomes, the formation of a holocentric line-like chromosome structure is still possible. One difference in their simulations appears to be the addition of centromeric nucleosome attraction? "Pairs of centromeric nucleosomes have a different attractive-repulsive force between each other, mimicking a selective sticky force". Could the authors comment further on this? Was this attraction necessary to simulate the line-like centromeric structure? Also, I am interested to know if the authors were to simulate a kinetochore (as in their previous study) would a groove structure be seen?

5: Supplementary movie 3 would not open for me.

REVIEWER COMMENTS

Reviewer #1

In this study the authors provide insights into the genome and nuclear organization of the holocentric plant *Chionographis japonica* using genomic and microscopic approaches.

Using antibodies that they generated against centromere and kinetochore components of *C. japonica* the authors confirmed the presence of holocentric chromosomes in this species with centromeres located over the entire length of mitotic chromosomes. They also found that, similar to some monocentric organisms, *C. japonica* centromeres in interphase overlap with a few large chromocenters. This is unlike other known holocentric organisms where centromere clusters have not been described. Using genomic approaches targeting CENH3 and other histone/DNA modifications they localized CENH3 to a few (~8) megabase-sized arrays per chromosome that are depleted in H3K4me2, flanked by H3K9me2 regions and enriched in CpG methylation. Again, these features resemble the organization of individual centromeres of the monocentric plant *A. thaliana* as well as the holocentric *R. pubera*. Using a bottom-up approach clustering the CENH3-enriched reads followed by DNA FISH experiments as a confirmation they determined that centromeres are enriched for two types of related satellites, Chio1 and Chio2. Interestingly, on mitotic chromosomes staining for the H3K9me2 mark (and late-replicating regions) resembles centromere staining. Finally, using polymer simulation approaches the authors propose a model underlying the structural dynamics of megabase-sized centromeric units clustered into chromocenters in interphase to the formation of a “centromere line” in mitotic chromosomes.

The study is technically sound, the writing is clear and the conclusions follow clearly from the results. It is also interesting to note that holocentromeres can consist of only a few large centromere units, each of which resembles a single monocentromere in other plants. I only have two minor comments.

RESPONSE: Many thanks for your helpful comments on how to improve the manuscript.

Comment 1:

Figure 2: The CENH3 and MIS12 immunosignals do not form the characteristic lines along the chromosomal surface as in Figure 1 and S3c? It seems to be rather telophase than anaphase as indicated?

RESPONSE:

We agree that Figure 2a shows a mitotic telophase cell instead of anaphase. We have corrected the Figure 2 legend to “mitotic telophase chromosomes” (Page 37, Line 12).

Comment 2:

Figure 6: The resemblance between the H3K9me2 and centromere immunosignals are interesting. How do the authors reconcile the fact that H3K9me2 is rather enriched in pericentric regions than at the centromere core?

RESPONSE:

The genome-wide H3K9me2 ChIPseq data shown in the H3K9me2 track/plot of Figures 5a and 5b indicates an accumulation of dimethylated H3K9 in the pericentric regions. Such a H3K9me2 enrichment at pericentromeric regions rather than in the centromere core is conserved from the monocentric e.g. *Arabidopsis thaliana* (see Figure 4A in (Naish et al., 2021)) to the holocentric species e.g. *Rhynchospora pubera* (see Figure 4C in (Hofstatter et al., 2022)).

However, during mitosis, chromatin folding/ condensation brings the pericentromeric regions of a chromosome close to each other resulting in a line-like pericentromere adjacent to the line-like centromere (Figure 6b). The resolution of microscopy is not sufficient to discriminate between the centromere and the pericentromere of a mitotic metaphase chromosome.

Reviewer #2 (Remarks to the Author):

The manuscript by Kuo et al. describes the unique holocentromeres of *Chionographis japonica*, which consist of ~9 distinct megabase-sized arrays of 23 and 28 bp tandem repeats per chromosome that assemble into a linear kinetochore. The authors also assemble much of the *C. japonicus* genome and map cenH3, H3K4 me2 and H3K9me2 nucleosomes and DNA methylation to address the overall genome architecture. They modify an earlier model of holocentromere condensation of many small centromeric units to account for how a small number of large centromere units can assemble into a linear kinetochore. This paper will be of interest to the fields of centromere biology and chromosomal structure not only because it is another unique example of independently derived holocentromeres but because it helps illuminate how a chromosome might evolve from a monocentromere to a metapolycentric centromere to a holocentromere. The writing is mostly clear, but I have a few mostly minor comments.

RESPONSE: Many thanks for your helpful comments on how to improve the manuscript.

Comment 1a:

On page 6, 2nd paragraph the authors discuss the number of chromocenters and conclude there are 2.8 per chromosome. Although it is easy to do the math, they

don't mention here or in Table S1 how many chromosomes *C. japonica* has until the next page, whereas it would be useful to mention it here. Is the chromosome number known from some earlier cytology (presumably the Tanaka references), or was it deduced from the genome assembly? The former seems more likely, since *C. japonica* was already known to be holocentric. A reference to the earlier work would be appropriate.

RESPONSE

Yes, the chromosome number of *C. japonica* of $2n = 24$ was already reported by (Tanaka and Tanaka, 1977; Tanaka and Tanaka, 1979), and our genome assembly showed the same result. As suggested, we added the chromosome number of *C. japonica* already in this paragraph with the two Tanaka references (Page 6, Line 12-13) and add this information to Supplementary Table 1 (Page 43, Line 20). Now it reads: "Thus, considering a diploid chromosome number of 24 (Tanaka and Tanaka, 1977; Tanaka and Tanaka, 1979), *C. japonica* forms, on average, as few as 2.8 CENH3-positive chromocenters per chromosome."

Comment 1b

Although they address the clustering of centromere units into chromocenters in the later modeling section and discussion, it would be helpful to point it out on page 8 where they determine that number of centromere units per chromosome is ~9, not 2.8.

RESPONSE

We agree and added the following sentence (Page 8, Line 10-11): "Thus, on average, ~3 centromere units are present in each chromocenter at interphase".

Comment 2:

At the bottom of page 6-top of page7, the authors did not detect H2AT120ph, and present in Fig S6 three H2A sequences, two with alanine in place of threonine at position 120, which cannot be phosphorylated, and one with serine, which possibly could be phosphorylated. Do the authors have information on the transcript abundance of these three genes or any other H2A genes in dividing (or non-dividing) tissues that might support the lack of a phosphorylatable H2A? This is not essential, but would be interesting if the information is available.

RESPONSE

As it is not known whether the antibody against H2AT120ph can also recognize H2AS120ph, the data about expression level of the two H2A variants could not explain the absence H2AT120ph immunostaining signal. We nevertheless quantified the expression level of the H2A variants H2ASer120 and H2AAla120 in the root transcriptome of *C. japonica*. The expression of the phosphorylatable H2ASer120

and non-phosphorylatable H2A*A/a*120 variants are similarly low, TPM (transcripts per million) of 0.059 for H2A*Ser*120 and 0.062 for H2A*A/a*120.

Comment 3:

On page 8 line 25, the authors specify nine CENH3 regions, but omit to say that this is per chromosome.

RESPONSE

We added “per chromosome” to the sentence (Page 8, Line 30-31). Now it reads: “Naturally extended pachytene chromosomes showed, on average, nine evenly spaced distinct scattered CENH3- and Chio1-positive centromere units colocalizing with knob-like chromatin structures per chromosome (Fig. 3g, Supplementary Fig. 11a-c).”

Comment 4:

In Figure 3e, the authors show that there are a nearly equal number of interspersed ‘forward’ and ‘reverse’ blocks of Chio1/Chio2 repeats. This seems unusual, and suggests considerable rearrangement of tandem sequences. The authors mention in the discussion (p. 17) the bridge-breakage-fusion model that might account for inverted sequences, but the authors might choose to elaborate a little on how they imagine this accounting for the distribution of forward and reverse blocks of satellites.

RESPONSE

Many thanks for your advice. We have rethought our original idea regarding the involvement of bridge-breakage-fusion events in the eccDNA formation and removed this speculative idea from the manuscript (Page 17, Line 8).

Comment 5:

In the legend to Figure 3g (p. 33), the authors refer to arrows but I cannot see any arrows. Please correct the legend or the figure.

RESPONSE

We added the missing arrows.

Comment 6:

In Figure 5b and 5d the x-axis is marked with TSS and TES, which makes sense for genes (blue) and possibly for the transposons (black) but not so much for Chio 1&2 (green). Perhaps these points can be labeled 'start' and 'end' instead of TSS and TES, or there can be an explanation in the legend.

RESPONSE

We corrected TSS and TES to start and end in the Figure 5b and 5d.

Comment 7:

In Figure S12, the legend says Patterns 1-3 correspond to early, mid and late replication, respectively, but in Figure S12b the 'late' pattern 3 shows up at three hours while the 'early' pattern barely shows up at 12 hours and isn't abundant until 24 hours. It appears that the colors for patterns 1 and 3 may have been switched in the inset legend in 12b, or else I don't understand what is being shown. Please correct or clarify.

RESPONSE

We would like to explain that the hours in Figure S12b (now Figure S13b) indicate the recovery time after the EdU incorporation, and the number of mitotic metaphases showing different EdU-labeling patterns was counted. The pattern observed after the shortest recovery time (3 hr), Pattern III, refers to an EdU incorporation at late S phase and the one obtained after the longest recovery time (24 hr), Pattern I, refers to an incorporation at early S phase. Consequently, we observed 3 and 6 hours after the EdU pulse, only metaphases with Pattern III (late replication pattern) while the number of Patterns II and I increased with increasing recovery time after the EdU pulse, representing mid and early replication patterns, respectively (please see the schemata below).

Comment 8:

In movie #2, there appear to be ~6 green blobs that vary in size but seem large for

telomeres and many fewer than I would expect for 2 ends x 24 chromosomes. These also do not look like clustered telomeres where several chromosomes come together, but rather like ~6 grooveless green chromosomes. Also, the purple Chio1 appears to be on several chromosomes, but not on all, and does not come together in a line, though the legend says this is metaphase. Please explain what is being shown. The legend merely says that telomeric DNA is green and Chio1 is purple.

RESPONSE

We are sorry for the misleading legend. The Supplementary Movie 2 shows a single mitotic metaphase chromosome of *C. japonica* labeled with telomere DNA (in green) and Chio1 (in purple) probes. The telomere signals are at the chromosome ends, and the Chio1 clusters which are closely associated to each other form a line-like holocentromere from telomere to telomere. At the end of the movie, the surface rendering shows a smooth surface of metaphase chromosomes. Unlike the holocentric plants *L. elegans* and *R. pubera* (Wanner et al. 2015), no chromosome groove exists in *C. japonica*.

We corrected the legend following (Page 44, Line 14): “A single mitotic metaphase chromosome of *C. japonica* labeled with telomere (in green) and Chio 1 (in purple) probes. The telomere signals are at the chromosome ends, and the Chio1 clusters closely associated to each other, form a line-like holocentromere from telomere to telomere. At the end of the movie, the surface rendering shows a smooth surface of the metaphase chromosome. Unlike the holocentric plants *L. elegans* and *R. pubera* (Wanner et al. 2015), a chromosome groove was not detected in *C. japonica*. Rendering of 3D-SIM image stacks was performed using Imaris 9.7.

Comment 9:

The file for movie #3 is corrupt and I could not open it.

RESPONSE

We are sorry for the inconvenience, the movie 3 is repaired.

Reviewer #3 (Remarks to the Author):

Review for Kuo et al., “Plasticity in centromere organization: Holocentromeres can consist of merely a few megabase-sized satellite arrays”.

The Houben lab specialise in investigating chromosome structure and function in plants and are particularly interested in centromere organisation, regulation, and evolution. Their recent Cell paper focused on holocentromeres. Unlike most species

that have a single monocentromere visualised as a primary constriction in mitosis, holocentric species form centromeres (100s) along the entire chromosome length (Hoffstatter et al 2022). This current manuscript (Kuo et al) further explores holocentromeres by focusing on another holocentric plant, the lilioid *Chionographis japonica*. Kuo et al assembled the reference genome for *C. japonica* and describe its centromere and epi-genome organisation. This study is specifically interesting as *C. japonica* has only a few (7-11) evenly-spaced megabase-sized centromeres which are CENH3 positive, far fewer and much larger than all other holocentric species described thus far. Interestingly in many of their analyses these holocentromeres function more like monocentromeric than holocentric species with 100s of centromeres along the chromosome, for example *C. japonica* form centromere clusters in chromocenters at interphase and the *C. japonica* genome is organised into distinct eu- and heterochromatin domains, similar to monocentric species. However, at the same time in *C. japonica*, like other holocentromeric species, microtubule attachment and kinetochore proteins colocalise with CENH3 positive regions along the length of the chromosome. These data very nicely demonstrate the plasticity of holocentric chromosome organisation, and the authors provide a great discussion on holocentromere evolution and where *C. japonica* might fit within this. Overall, this is a nicely conducted and compelling study of a novel holocentromere which will broaden our mainly monocentric-based understanding of centromere identification and function.

I have some minor points for the authors to address that should clarify a few questions that came from reviewing this manuscript:

RESPONSE: Many thanks for your helpful comments on how to improve the manuscript.

Comment 1:

1: The authors have previously shown that in other holocentric species *L. elegans* and *R. pubera* the CENH3- positive centromere forms a longitudinal groove at metaphase. One specifically interesting result with *C. japonica* is that this groove structure is absent.

Do the authors think the absence of a central groove is significant? Could this be related to the absence of H2AT120ph in *C. japonica*?

RESPONSE

According to the current knowledge, we can't conclude on a correlation between the loss of centromeric H2AT120ph and the absence of a centromere groove. For example, the metaphase groove is less defined in the holocentric *L. nivea* with smaller chromosomes (Nagaki et al., 2005), not detectable in *L. luzuloides* but clearly detectable in *L. elegans* (Heckmann et al., 2011), meaning a longitudinal groove is not always present in the H2AT120ph-positive metaphase chromosomes of holocentric *Luzula* species. We assume that the distinct H2AT120ph-positive centromeric groove in holocentric *L. elegans* and *R. pubera* is likely a structural accommodation for the stability of large-sized mitotic chromosomes. On the other hand, no centromeric H2AT120ph was found in the holocentric subgenus *Cuscuta*

species, while it is present in monocentric *Cuscuta* species (Neumann et al., 2020). Also, holocentric *Cuscuta* chromosomes show no groove.

Comment 2:

2: The authors use Supplementary movie 2 to support their finding of a lack of longitudinal groove. I don't find this movie particularly helpful in presenting this. Perhaps a surface rendering of SIM image stacks of one/a few chromosomes (as they have presented previously in Fig.1 Marques et al 2015) would more clearly display the lack of grooves.

RESPONSE

We are sorry that the initial legend of this movie was misleading. The Supplementary Movie 2 shows a single mitotic metaphase chromosome of *C. japonica* labeled with telomere DNA (in green) and Chio1 (in purple) probes. The telomere signals are at the chromosome ends, and the Chio1 clusters are close together, forming a line-like holocentromere from telomere to telomere. At the end of the movie, the surface rendering shows a smooth surface of metaphase chromosomes. Unlike in the holocentric plants *L. elegans* and *R. pubera* (Wanner et al. 2015), no chromosome groove exists.

We corrected the legend following (Page 44, Line 14): "A single mitotic metaphase chromosome of *C. japonica* labeled with telomere (in green) and Chio1 (in purple) probes. The telomere signals are at the chromosome ends, and the Chio1 clusters closely associated to each other, form a line-like holocentromere from telomere to telomere. At the end of the movie, the surface rendering shows a smooth surface of the metaphase chromosome. Unlike the holocentric plants *L. elegans* and *R. pubera* (Wanner et al. 2015), a chromosome groove was not detected in *C. japonica*. Rendering of 3D-SIM image stacks was performed using Imaris 9.7.

Comment 3:

It is also not clear to me in this movie why the purple signal of the Chio 1 satellite is not present on all chromosomes as Figure 3C and 3H indicate Chio 1 repeats are found on all chromosomes.

RESPONSE

Probably our legend of Supplementary Movie 2 was misleading. This movie shows a single metaphase chromosome of *C. japonica* labeled with telomere (in green) at the chromosome ends and the Chio 1 clusters (in purple) which were brought close together into a line-like holocentromere from telomere to telomere.

Comment 4a:

3: I am particularly interested in how the authors have mapped CENH3 ChIP seq reads to specific Chio 1 and Chio 2 repeat regions in *C. japonica*. Can the authors provide more explanation as to how these repetitive short read sequences were mapped to specific regions on each of the 12 different *C. japonica* chromosomes. My understanding is that as these are repetitive reads, one cannot be certain the specific location of each individual read, therefore these must have been randomly distributed.

RESPONSE

The mapping of the CENH3 ChIPseq to specific Chio1 and Chio2 repeat regions was possible due to the unique organization of the centromere units in *C. japonica*. Because although the holocentromere of *C. japonica* is Chio satellite DNA-based, the arrangement of Chio repeats is highly dynamic due to the forward and reverse orientation of Chio1 and Chio2 blocks and the different sizes of individual Chio1 and Chio2 blocks. To demonstrate this, we randomly selected a 6-kb and a 1-kb region from a centromere unit of chromosome 2 for the dot plot analysis. Despite the tandemly arrayed nature of the Chio repeats, the orientation of Chio arrays changed seven times in the 6-kb region (Figure 1a) and two times in the 1-kb region (Figure 1b). Also, the size differs between the individual arrays. Thus, the unique organization of Chio1 and Chio2 blocks of each centromere unit enabled us to assemble the holocentromeres of *C. japonica* using long PacBio HiFi reads.

Figure 1 Dot plot analysis of the randomly selected (a) 6-kb and (b) 1-kb centromeric regions of chromosome 2 of *C. japonica*. The blue and green represent the sequence similarity in the same and opposite orientations, respectively.

Additionally, the monomer sizes of the Chio1 and Chio2 repeats are short, only 23 and 28 bp in length, respectively. Thus, each 150-bp Illumina ChIPseq read contains several Chio repeat units, even with some insertions/deletions between or within Chio monomers. In other species like *Arabidopsis thaliana* and *Rhynchospora pubera*, one

short Illumina sequence read represents only one or part of a single centromeric repeat monomer in length of 170 – 180 bp, which makes it technically difficult in CENH3-ChIPseq mapping.

To demonstrate this, we selected single-end 150-bp CENH3-ChIPseq reads containing at least 2-3 perfect-matching Chio1 monomers, meaning they are CENH3-ChIPseq reads with higher sequence similarity. Dot plot analysis (Figure 2a) and multiple alignments (Figure 2b) between these reads showed that they are similar but not identical. Thus, the sequence differences and the arrangement of Chio repeats allowed the mapping of CENH3-ChIPseq reads to the PacBio HiFi-based genome assembly of *C. japonica*.

(a)

(b)

Figure 2 Dot plot analysis and multiple sequence alignments between single-end 150-bp CENH3-ChIPseq Illumina reads. The selected reads contain at least 2-3 perfect-matching Chio1 repeat monomers, meaning reads with higher sequence similarity. Nevertheless, (a) dot plot analysis and (b) multiple alignment show that they are not identical despite their high sequence similarity.

Besides, to exclude the bias caused by the ChIPseq mapping in the multi-mapping mode, we filtered out all the multi-mapped reads from the Bowtie2 outputs of both Input and CENH3-IP samples using Sambamba (Tarasov et al., 2015). The \log_2 ChIP/Input ratio in the uni-mapping mode was calculated using bamCompare for the

1 kb window size. The resulting distribution patterns of CENH3-ChIPseq analyzed using the uni-mapping mode are almost identical as using the multi-mapping mode (Figure 3).

Figure 3 Comparison of the CENH3- and H3K9me2-ChIPseq profiles generated using the multi- and uni-mapping modes. The uni-mapping patterns (Uni, tracks in magenta) were generated by filtering out all the multi-mapped reads in the Bowtie2 outputs using the multi-mapping mode (Multi, tracks in blue). The \log_2 CHIP/Input ratio was calculated for the 1 kb window size. The CENH3- and H3K9me2-ChIPseq profiles analyzed using the multi-mapping mode in the (a) Chromosome 2 and Chromosome 4 of *C. japonica* are corresponding to the tracks in the Figure 4, 5, and Figure S12 of the manuscript.

Comment 4b:

Similarly, with the H3K9me2 reads that map to repetitive Chio 1 or 2 sequence, it is not clear to me how the authors have been able to assign these to specific repeat

regions. The authors state that “part of the centromere units were depleted of H3K9me2” but I am unclear as to how they have been able to know which exact repeat region these reads were derived from.

RESPONSE

Similar to the response above, the short Chio repeat monomer and dynamic organization enabled the mapping of H3K9me2-ChIPseq reads to the assembled genome. Especially, the enrichment of H3K9me2 was detected at the flanking regions of centromere units, not in Chio-rich centromere units. We observed only some minor differences between the distribution patterns of H3K9me2-ChIPseq using the two mapping modes (Figure 3). Nevertheless, the H3K9me2 enrichment showed similar tendency of accumulating at pericentric regions, and the results from the uni-mapping mode showed even higher pericentric enrichment compared to those using the multi-mapping mode. We added the Figure 3 in the response letter as the Supplementary Figure 9 in the manuscript.

Comment 5:

4: The authors have previously used polymer-based modelling to propose a mechanism for how holocentric chromosomes form a line-like centromeric structure in metaphase (Camara et al 2021). In this manuscript they use the same approach to describe how, with only a few centromere units along C. Japonica chromosomes, the formation of a holocentric line-like chromosome structure is still possible. One difference in their simulations appears to be the addition of centromeric nucleosome attraction? “Pairs of centromeric nucleosomes have a different attractive-repulsive force between each other, mimicking a selective sticky force”. Could the authors comment further on this? Was this attraction necessary to simulate the line-like centromeric structure? Also, I am interested to know if the authors were to simulate a kinetochore (as in their previous study) would a groove structure be seen?

RESPONSE

Yes, this attraction between centromeric nucleosomes was also applied during the condensation of the modelled chromosome by loop extrusion. Without this attraction, the centromeric units form a long line of centromeric nucleosomes interspaced by very short chromatin loops. This long line hinders the approach of condensed large non-centromeric regions, and the entire model shows separated volumes of condensed chromatin and the centromeric units as linkers between them. We added two sentences in the manuscript to make this point clearer (Page 12, lines 10-15). It reads, “Throughout this process, the specific attraction between centromeric nucleosomes persisted, preventing the centromeric units from creating a long line with short chromatin loops in between. Thus, chromatin loops accumulated in the vicinity of the chromocenters, while chromocenters remained condensed by the applied attraction force, resulting in a non-uniformly condensed chromosome (Fig. 7b-c, Suppl. Movie 3).”

Prompted by the reviewer, we simulated the kinetochore using the same conditions as previously proposed for a general holocentric model, with the simple assumption

that centromeric nucleosomes align uniformly along the kinetochore complex, which was simulated with the same size. First 20,000,000 time steps were needed to equilibrate the long regions of chromatin in between centromeric units closer to the simulated kinetochore complex before the extrusion loop starts like in the general holocentric model. After 37,000 block steps of loop extrusions, we compared the conformation of a *Chionographis*-type chromosome with a general holocentric chromosome in the presence of the kinetochore (Figure 4). Like in the general holocentric model, the kinetochore poses a constriction along the *Chionographis* chromosome model, but a groove is not as evident (Figure 4a). We attribute this phenomenon to the different distribution of chromatin loops. In the general model, loop extruders clearly align at the side of the centromeric line (Figure 4b), while in the *Chionographis*-type model, loop extruders cluster in the center of each chromatin region between centromeric units (Figure 4a).

Figure 4 Two models of a single chromatid in the presence of a kinetochore complex, (a) *Chionographis*-type chromosome and (b) general holocentric chromosome. The kinetochore complex is represented by the grid of light blue beads. Red beads represent centromeric nucleosomes, and yellow beads nucleosomes bound by a loop extruder (representing condensing complexes). Three different views are shown for each model: on the left, with a visible kinetochore complex; in the middle, the kinetochore complex is hidden, and the line of centromeric nucleosomes is evident; on the right, a slice of the chromosome shows the distribution of loop extruders.

Comment 6:

5: Supplementary movie 3 would not open for me.

RESPONSE

We are sorry for the inconvenience, the movie 3 file is repaired.

References

- Heckmann, S., Schroeder-Reiter, E., Kumke, K., Ma, L., Nagaki, K., Murata, M., Wanner, G., and Houben, A. (2011). Holocentric Chromosomes of *Luzula elegans* Are Characterized by a Longitudinal Centromere Groove, Chromosome Bending, and a Terminal Nucleolus Organizer Region. *Cytogenet Genome Res* 134:220-228.
- Hofstatter, P.G., Thangavel, G., Lux, T., Neumann, P., Vondrak, T., Novak, P., Zhang, M., Costa, L., Castellani, M., and Scott, A. (2022). Repeat-based holocentromeres influence genome architecture and karyotype evolution. *Cell* 185:3153-3168. e3118.
- Nagaki, K., Kashihara, K., and Murata, M. (2005). Visualization of Diffuse Centromeres with Centromere-Specific Histone H3 in the Holocentric Plant Luzula nivea. *The Plant Cell* 17:1886.
- Naish, M., Alonge, M., Wlodzimierz, P., Tock, A.J., Abramson, B.W., Schmücker, A., Mandáková, T., Jamge, B., Lambing, C., Kuo, P., et al. (2021). The genetic and epigenetic landscape of the *Arabidopsis* centromeres. *Science* 374.
- Neumann, P., Oliveira, L., Cizkova, J., Jang, T.S., Klemme, S., Novak, P., Stelmach, K., Koblizkova, A., Dolezel, J., and Macas, J. (2020). Impact of parasitic lifestyle and different types of centromere organization on chromosome and genome evolution in the plant genus *Cuscuta*. *New Phytol.*
- Tanaka, N., and Tanaka, N. (1977). Chromosome studies in *Chionographis* (Liliaceae) .1. Holokinetic nature of chromosomes in *Chionographis japonica* Maxim. *Cytologia* 42:753-763.
- Tanaka, N., and Tanaka, N. (1979). Chromosome-Studies in *Chionographis* (Liliaceae) .2. Morphological-Characteristics of the Somatic Chromosomes of 4 Japanese Members. *Cytologia* 44:935-949.
- Tarasov, A., Vilella, A.J., Cuppen, E., Nijman, I.J., and Prins, P. (2015). Sambamba: fast processing of NGS alignment formats. *Bioinformatics* 31:2032-2034.

REVIEWERS' COMMENTS

Reviewer #2 (Remarks to the Author):

In the revised manuscript by Kuo et al., the authors have replied to my concerns satisfactorily and appear to have also done a good job of responding to the other reviewers. This paper will be an interesting addition to the holocentromere literature.

Reviewer #3 (Remarks to the Author):

This is a revised manuscript from Kuo et al., "Plasticity in centromere organization: Holocentromeres can consist of merely a few megabase-sized satellite arrays".

Kuo et al have assembled the reference genome for the holocentric plant *C. japonica* and describe its centromere and epi-genome organisation. This study is specifically interesting as *C. japonica* has only a few (7-11) evenly-spaced megabase-sized centromeres which are CENH3 positive, far fewer and much larger than all other holocentrics described thus far. Intriguingly *C. japonica* holocentromeres have properties of both monocentromeres and holocentromeres demonstrating the plasticity of holocentric chromosome organisation whilst also revealing how monocentromeres could evolve to holocentromeres.

I thank the authors for addressing all my comments. In particular, I hadn't understood how they had specifically mapped CENH3 & H3K9me2 ChIPseq reads to the Chio 1 & 2 repeats. The authors have provided a detailed explanation and added more information and another figure to the manuscript. Additionally, I was intrigued to know if they had simulated a kinetochore in their polymer simulations and how this might affect the centromere structure. The authors have completed these additional simulations and this data also supports their experimental findings of a lack of longitudinal groove at metaphase.

Overall, this is a nicely conducted and compelling study of a novel holocentromere which will broaden our mainly monocentric-based understanding of centromere identification, structure and function making it worthy of publication in Nat Comm.